# Multi-Level Perceptions on Higher Education Development for Sanitation and Hygiene Management in Nigeria

Peter Emmanuel Cookey [1,2,3,*], Mayowa Abiodun Peter-Cookey [4], Iloma Unwobuesor Richard [2] and Damir Brdanovic [3]

1   Institute of Geosciences and Environmental Management (IGEM), Rivers State University, Port Harcourt 500101, Nigeria
2   School of Environmental Health, Rivers State College of Health Sciences and Technology Management, Port Harcourt 500272, Nigeria; richardiloma528@gmail.com
3   Water Supply, Sanitation and Environmental Engineering Department, IHE Delft Institute for Water Education, 2611 AX Delft, The Netherlands; d.brdjanovic@un-ihe.org
4   Research and Development Department, EarthWatch Research Institute, 3 David Nna Layout, Rumuogba, Port Harcourt 500102, Nigeria; live4mayo@gmail.com
*   Correspondence: cookeypeter@gmail.com

**Abstract:** Providing safely managed sanitation/hygiene requires key competencies for education, training, service delivery, enterprise development and management, product and infrastructure design and development, construction and installation, governance, financing, research, etc. These strategic sanitation capacities will have to be built from higher education's academic and professional programmes structured and designed to produce skilled and knowledgeable professionals and practitioners. This study aims to investigate the quality of the sanitation/hygiene management content of Environmental Health programmes; the adequacy of the existing central curricula; and the perception of environmental health sanitation/hygiene-trained professionals to determine the next phase for building knowledge and capacity of sanitation professionals in Nigeria through higher education institutions. A multi-level mixed method concurrent study was used for sampling and data collection with a multi-level perception analysis to examine the perceptions of students, lecturers and graduate alumni, as well as employers/supervisors and clients/service users of sanitation/hygiene-related graduates. The main findings of this paper show a limited understanding of the concepts of contemporary issues of sanitation/hygiene management like sustainable sanitation, citywide inclusive sanitation, regenerative sanitation, circular bioeconomy, etc.; the central/national teaching and curricula were found to be seriously out of date; and clients/service users were dissatisfied with the skills and knowledge levels of the graduates. The study concludes by recommending a national sanitation management higher education pathway to guide the provision of integrated sanitation/hygiene management education at higher education institutions (HEIs), to build an effective, efficient, competent and sufficient workforce for the country.

**Keywords:** skills; knowledge and competency; multi-level perception; sanitation/hygiene management; higher education institutions; environmental health technician/technology





## 1. Introduction

Higher education (HE) in the field of sanitation and hygiene (SH) management is critical to access and delivery of safe, resilient and affordable services and systems, particularly in the developing countries of sub-Saharan Africa. This will involve teaching, learning, research and innovative contributions to society with the potential to drive economic and social transformation [1], to provide skilled and knowledgeable human capital, as well as leaders and managers that will drive the journey towards the SDG 6 targets on sanitation and hygiene [2–5]. However, the field of sanitation and hygiene (SH) management is scantily represented in higher institutions (HEIs) of most African countries in spite of the fact

that these institutions are supposed to function as a hub for developing human resources for key fields and sectors relevant to the industrial and socioeconomic development of their societies [1,4,5]. Bloom et al. [6] suggest that when countries have a high number of HE graduates in their labour force, they are more productive and able to easily flow with new techniques and innovation, which could help improve the poor status of sanitation and hygiene in most of these countries.

For example, in Nigeria, water management is well represented in HEIs (all Universities in Nigeria have one form of water-related course from bachelor to PhD levels), and the country has made great progress whereby water resources and supply goals and targets are almost achieved for its over 200 million population. Meanwhile, sanitation/hygiene management is minimally represented in the HEIs, and only as part of other related programmes, which is indicative of the current situation whereby SH is way below the target with the country ranking third most backward [7]. Even the federal government's desperate state-of-emergency declaration and 13-year National Action Plan for the Revitalization of the Water, Sanitation and Hygiene (WASH) sector (2018) noted that limited knowledge and capacity in the sector is a huge hindrance to progress [8]. In fact, many sub-Saharan African countries still lag behind on the SDG 2030 sanitation/hygiene targets, particularly in urban centres, and there is a lack of labour and up-to-date knowledge in the sector.

The pursuit of safely managed SH in countries like Nigeria requires key skills and competencies for education, training, service delivery, enterprise development and management, product and infrastructure design and development, construction and installation, governance, financing, research, etc. These strategic sanitation capacities will have to be built from HEI academic and professional programmes structured and designed to produce skilled and knowledgeable professionals and practitioners. It is expedient to have a competent, efficient, and effective workforce; otherwise, the investments to improve sanitation will go down the drain as overwhelmed and under-equipped sanitation managers fail to perform their responsibilities adequately and appropriately. Therefore, an upgrade in the capacity and knowledge of sanitation professionals (public and private) is crucial for the WASH National Action Plan to be effective and it is really critical to the implementation of policies, legislation, standards, strategies and programmes towards meeting the SDGs in general and sanitation targets (e.g., end open defecation, increase access, improve service delivery, etc.).

A national capacity and knowledge development programme at the HE level for SH management could be a means to harness all resources to develop and implement training and education structures as well as the enabling environment to increase and upgrade the competence and quality of sanitation managers in Nigeria. However, any programme must take cognizance of upgrading the training of Environmental Health Officers (EHOs) and community health officers (CHOs) as primary stakeholders in the pursuit of improving sanitation across all administrative levels of government. Environmental health professionals are the regulators and operators of SH management and are specifically mandated by law to oversee sanitation practices in Nigeria (and other West African countries), while community health professionals are mandated with sanitation and hygiene training at community primary healthcare centres. However, the key educational programmes that particularly train these sanitation management officers have received minimal knowledge and capacity-building interventions and/or provisions in over 50 years. The training and education of these labour groups are structured within Colleges of Health (HEIs) across the 36 states of the federation and the Federal Capital Territory (FCT) (and recently bachelor and postgraduate degrees in some universities) with a central curriculum developed by the Environmental Health Council of Nigeria (EHCON) and the West African Health Examination Board (WAHEB) for EHOs and the Community Health Practitioners Registration Board of Nigeria for CHOs. Meanwhile, the federal government through the Federal Ministry of Water Resources signed a memorandum of understanding with the world-renowned IHE Institute of Water Education in the Netherlands to transfer Master's level programmes in sanitation to six Nigerian universities, among other things, so as to improve the capacity

and availability of labour to support the urgent drive of the government (which is yet to materialise). However, since the focus is postgraduate level, it still does not cover the colleges where the EHOs and CHOs who manage SH across state and federal levels are trained or the possibility of Bachelor's-level education.

This has led to mostly outdated curricula, inadequate teaching personnel, minimal focus research, and dissatisfaction among clients and service users of SH provisions. Thus, this particular study aims to investigate the quality of the SH management content of Environmental Health programmes; the adequacy of the existing central curricula, especially as it regards the current challenges of the era; and the perception of environmental health sanitation/hygiene-trained professionals and users of their services so as to determine the next phase for building knowledge and capacity of sanitation professionals in Nigeria through HEIs. SH management, in view of this study, is separate from solid waste management and focuses on the management of by-products of human and animal digestion like excreta and urine.

## 2. Methods

An exploratory survey was conducted using multi-level perception analysis to examine the perceptions of students, lecturers and graduate alumni of the Environmental Health (EHT) programmes of a College of Health and Technology in South-South, Nigeria, as well as employers/supervisors and clients/service-users of graduates [9]. This is due to the fact that the perceptions of multi-level stakeholders were sought to determine an overall perception concerning SH education in Nigeria. A multi-level mixed method concurrent study was used for sampling and data collection with the instruments of self-administered semi-structured questionnaires containing closed and open-ended questions for all respondents and interviews. This was due to the fact that in some cases there was no structured access to participants and we had to use purposive sampling, while in other cases like students and lecturers, we used a random selection from the school list.

The self-administered semi-structured questionnaires were administered in three groups based on how the test instrument was taken. Group A took the test before filling out the questionnaires; Group B took the test after filling out the questionnaire; and Group C comprised members from Groups A and B who agreed to fill out the questionnaire a second time. Observation and review of the literature were also for in-depth comprehension and corroboration. The study area of this survey was selected because it was the first School of Health Technology to transition into a full-fledged monotechnic college in the country and has now been upgraded to a polytechnic status. The environmental health field of study was chosen because its graduates (i.e., the Environmental Health Officers) are the ones officially mandated in the Nigerian Constitution [10] and the law [11] to manage and regulate sanitation issues in the country. If the battle for sanitation will be won in Nigeria, these soldiers will need to be equipped with appropriate and up-to-date extensive knowledge skills and capabilities.

### 2.1. Sampling

A mix of random and purposeful sampling was used on the various group levels of respondents [12–16] to select representative samples (where possible) and information-rich samples (when not possible), to enable in-depth study and convergence of results [16–21] and to ensure that participants for quantitative and qualitative investigations come from the same pool [14,22–24]. Students were randomly selected per study year and programme through a blind selection of matriculation numbers and then they were given the choice of participation. A total of 122 students participated in the survey (see Table 1) (i.e., 47.5%); 15 lecturers took part (i.e., 45%) with 13 of them being core sanitation-related lecturers. The graduate alumni were randomly selected from an archival list of more than 200 graduates, but only 86 participated (41%), and a total of 32 participated in the survey, while about 25 service users completed the process. All respondents signed consent letters.

**Table 1.** Respondents' sociodemographic profile.

| Characteristic | Description | Students | Graduates | Lecturers | Employers/ Supervisors | Clients/ Service Users | Sum |
|---|---|---|---|---|---|---|---|
| | | % | % | % | % | % | % |
| Gender | Female | 42 | 45 | 20 | 34 | 52 | 42 |
| | Male | 58 | 55 | 80 | 66 | 48 | 58 |
| Age (years) | 18–27 | 66 | 24 | - | 6 | 24 | 39 |
| | 27–37 | 27 | 36 | 40 | 19 | 20 | 29 |
| | 37–47 | 7 | 21 | 33 | 36 | 28 | 18 |
| | 47–57 | - | 9 | 27 | 36 | 28 | 11 |
| | >58 | - | 9 | - | - | - | 3 |
| Educational status | Primary/Secondary | 40 | - | - | - | 16 | 19 |
| | Technical | 7 | - | - | - | - | 3 |
| | Vocational | - | - | - | - | - | - |
| | OND | 44 | 9 | - | 31 | 13 | 27 |
| | HND | 8 | 22 | - | 13 | 24 | 14 |
| | Bachelor's | - | 14 | - | 34 | 32 | 11 |
| | PGD | - | 14 | 27 | - | - | 6 |
| | Master's | - | 37 | 53 | 16 | 8 | 17 |
| | PhD | - | 4 | 20 | 6 | - | 3 |
| | Other/ Professional | - | - | - | - | 8 | 1 |
| Employment status | Yes | 6 | 34 | 100 | 100 | - | 33 |
| | No | 79 | 35 | - | - | - | 49 |
| | Somewhat | 16 | 31 | - | - | - | 19 |
| Type of work establishment | Government Ministry | 2 | 17 | - | 25 | - | |
| | Government Agency | 1 | 5 | - | 25 | - | |
| | Higher Education/TVET Institution | 2 | 7 | - | 13 | - | |
| | Local Government | 2 | 14 | - | 9 | - | |
| | Sanitation-related Company | - | - | - | 19 | - | |
| | Non-governmental Organisation | - | 2 | - | - | - | |
| | Community-based Organisation | - | 1 | - | 3 | - | |
| | Informal/SME | 17 | 14 | - | - | - | |
| | Other | 2 | 7 | - | 6 | - | |
| | None | 75 | 35 | - | - | - | |
| Graduated from this same institution at some time | No | - | - | 27 | 34 | - | |
| | Yes | - | - | 53 | 66 | - | |
| | Not exactly | - | - | 20 | - | - | |
| Current programme (was) enrolled in | Environmental Health Technician (Certificate) | 34 | 21 | 67 | - | - | |
| | Environmental Health Technology (HND) | 66 | 79 | 100 | - | - | |

*2.2. Data Collection*

Perceptions of final-year students of the Environmental Health Technicians certificate programme and third-year Higher National Diploma (HND 1) and final-year (HND 2)

students of the Environmental Health Technology programme (equivalent to a Bachelor's degree) were surveyed to explore how they view the sanitation management content of the EHT programmes they are involved in. In addition, perceptions of lecturers from these two programmes who teach core sanitation (and related) courses as well as alumni who graduated from these programmes within the past 15–20 years were surveyed. For convergence, perceptions of employers/supervisors and users of services of these sanitation workers and professionals were sought. The students and lecturers were accessed through a lecturer who teaches in both programmes (who is also a co-author in this paper) while graduate alumni and employers/supervisors were sourced from a collation from the alumni archives and other sources such as the State Ministries of Environment and Health as well as the Waste Management Agency, the Rural Water Supply and Sanitation Agency and the Water Corporation (which also suggested service users/clients through convenient sampling). The survey employed the use of a mixed brew of questionnaires, interviews, observation and a desktop literature review to achieve credibility, trustworthiness and reliability through corroboration for triangulation [15,16,21,25–28].

*2.3. Test of Knowledge*

The survey also included a Test of Knowledge to determine the knowledge and understanding of the respondents on current sanitation management fundamentals and progress and was administered to only the students, lecturers and graduates. The Test of Knowledge was analysed based on how and if the questions were answered (and not on the number of correct answers) and on the number of persons that selected the same answers. There were three parts:

**Part One** had five multi-choice questions with five options to choose one or two answers only; the *Section 1* questions were about basic sanitation information such as *(i) sanitation is. . .; (ii) sanitation system is. . .; (iii) sanitation services are. . .; (iv) sanitation facilities include. . .; and (v) examples of non-sewer sanitation include. . ..* The *Section 2* contained five questions with open answers (e.g., *(i) mention two examples of onsite sanitation systems; (ii) mention four examples of sanitation behaviour; (iii) state four principles of sanitation management; (iv) state two differences between sewer and non-sewer sanitation; and (v) state the SDG 6 targets on sanitation*).

**Part Two** sought to determine respondents' knowledge of current and up-to-date sanitation management practices. It included 15 questions with open answers (*e.g., (i) What is sanitation technology?; (ii) What is decentralized sanitation?; (iii) What is the sanitation service chain?; (iv) What is faecal sludge management?; (v) What is sustainable sanitation?; and (vi) What is the shit-flow diagram?, etc.*).

Then, **Part Three** contained five open-answer questions to determine their knowledge level on sanitation resource recovery and reuse (*e.g., (i) mention four processes to recover sanitation materials; (ii) state four examples for reusing recovered sanitation materials; (iii) state four examples of sanitation behaviour change intervention frameworks; (iv) state three key global sanitation organisations; and (v) what is Nigeria's ranking in sanitation management across Africa and globally?*). There were 30 questions in all and participants had 30 min–2 h to take the test; depending on when they were taking the test (before or after filling out the questionnaire).

*2.4. Data Analysis*

A mixed data analysis design was applied at the analytical stage of the research to process quantitative and qualitative data, separately and in combination. Descriptive statistics (mean, frequency and percentage) were used for the analysis of quantitative and some qualitative data while thematic analysis were used for other qualitative (interviews, observations, literature review, etc.) data to find common themes and sub-themes. Some qualitative data were quantitized through frequency counts based on subthemes [14,25,27,29–34].

2.4.1. Multi-Level Perception Analysis Model

A multi-level perception analysis model was adapted to determine Overall Perception Estimations on the sanitation management content of the programmes and was then matched and triangulated. Overall perceptual estimations were matched and triangulated [14,25,27,29,31,33,35] using the exploratory multi-level perception rating analysis developed by Peter-Cookey and Janyam in the Prince of Songkla University, Hat Yai, Thailand [36–38]. Perception Estimations were determined using percentage MEAN and referred to combined scores of more than one sample group/category level. It was developed based on the principle of matching results from multiple raters to derive a single score in order to reduce bias and increase reliability. Equations were generated for each level of assessment. Five-point Likert scales were used to represent agreement, satisfaction and ratings.

The different respondent categories were designated as category levels (CLs) and they comprise students, graduates/alumni, lecturers, employers/supervisors and clients/service users. Context Areas (CA) are the different areas of concern considered in the survey such as satisfaction with value and expectations. Item/statement refers to the individual concerns raised in each Context Area. Groups refer to the respondents as they are separated based on the administration of the Test of Knowledge (ToK). Perception Estimations are the perception responses of the survey participants on individual concerns of the Cas. Overall Perception Estimation is the final score for each Context Area from the sum of each CL while Total Overall Perception Estimation is the sum for each Context Area for all the CLs.

Group Perception Estimations

Group Perception Estimations are based on the responses from each group in specific CLs per scale. First, perception responses for each Group (A, B and C) in CLs 1–3 (students, graduates/alumni and lecturers) concerning each item/statement in a specific Context Area were computed per scale (*e.g., Students Group A—Strongly Agree; Students Group B—Strongly Agree; Students Group C—Strongly Agree, etc.) and then total scores for all the items/statements in each Context Area of each group was derived by adding the responses on all the item/statement per scale. This was carried out separately for each group in each CL.*

$$
\begin{aligned}
\text{GPE-\#CA} &= \text{SGrA}isa1 + \text{Sgr}Aisb1 + \text{SgrA}isc1 + \text{Sgr}Aisd1 + \text{Sgr}Aisa1 + \text{Sgr}Aise1 = \textstyle\sum \\
\text{GPE-\#CA} &= \text{Sgr}Bisa1 + \text{Sgr}Bisb1 + \text{Sgr}Bisc1 + \text{Sgr}Bisd1 + \text{Sgr}Bisa1 + \text{Sgr}Bise1 = \textstyle\sum \\
\text{GPE-\#CA} &= \text{Sgr}Cisa1 + \text{Sgr}Cisb1 + \text{Sgr}Cisc1 + \text{Sgr}Cisd1 + \text{Sgr}Cisa1 + \text{Sgr}Cise1 = \textstyle\sum
\end{aligned}
\tag{1}
$$

where GPE-#CA stands for Group Perception Estimation on particular item/statement in a specific Context Area; GrA/B/C represents Group A, B, C; is a/b/c/d represents item/statement(a/b/c); while S represents students' perception per item/statement in Context Area, L represents lecturers' perception per item/statement in Context Area, GA represents graduates/alumni' perception per item/statement in Context Area, and 1 = strongly disagree; 2 = disagree; 3 = neutral; 4 = strongly agree; 5 = agree.

Overall Group Perception Estimations

Overall Group Perception Estimations are derived from summing up the responses of each group on specific CA in all the CLs per scale.

$$
\text{OGPE-\#CA} = \text{SgrA}(CA)1 + \text{Lgr}A(CA)1 + \text{GAGrA}(CA)1 = \textstyle\sum
\tag{2}
$$

Then, divide the total of each scale (e.g., strongly disagree) by the total number of responses in that CA and multiply by 100 to obtain a percentage mean for each scale in that CA.

$$
\text{OGPE-\#CA} = \text{SgrA}(CA)1 + \text{LGrB}(CA)1 + \text{GAGrC}(CA)1 = \frac{\sum(\text{SgrA}(CA)1 + \text{LGrB}(CA)1 + \text{GAGrC}(CA)1)}{\text{TNSCA}} \times 100
\tag{3}
$$

where OGPE-#CA stands for Overall Group Perception Estimation in a specific Context Area; CA represents Context Area; while S represents students' perception in context

area, L represents lecturers' perception in context area, GA represents graduates/alumni' perception in context area, TNSCA is total number of scores and 1 = strongly disagree; 2 = disagree; 3 = neutral; 4 = strongly agree; 5 = agree.

Overall Multi-Level Perception Estimations

Overall Multi-level Perception Estimations are based on the responses of survey participants in all CLs for all CAs. The steps began with collating the responses for each CL per scale in specific CAs. To do this, the perception responses for each item/statement (e.g., teaching methods) in a particular Context Area (e.g., value and expectations), the number of respondents in each CL that scored on a particular scale (e.g., agree/satisfied/outstanding) was computed separately and their percentages derived. The score for each item/statement was added across all the CLs to obtain a sum figure for individual concerns, divided by the total number of respondents and multiplied by 100.

$$
\begin{aligned}
\text{OMPE-\#CA} \quad &= \text{S(CA)1} + \text{L(CA)1} + \text{GA(CA)1} + \text{ES(CA)1} + \text{CS(CA)1} \\
&= \frac{\sum(\text{S(CA)1} + \text{L(CA)1} + \text{GA(CA)1} + \text{ES(CA)1} + \text{CS(CA)1})}{\text{TNR}} \times 100
\end{aligned}
\tag{4}
$$

where OMPE-#CA stands for Overall Multi-level Perception Estimation in a specific Context Area; CA represents Context Area; while S represents students' perception in context area, L represents lecturers' perception in context area, GA represents graduates/alumni' perception in context area, ES represents employers/supervisors' perception in context area, CS represents clients/service users' perception in Context Area, TNR is total number of respondents and 1 = strongly disagree; 2 = disagree; 3 = neutral; 4 = strongly agree; 5 = agree.

Afterwards, total scores for all the items/statements in each Context Area were derived by adding the responses on all the items/statements per scale. This was conducted separately for each CL. To determine the Overall Multi-level Perception for any Context Area (e.g., value and expectations), the sums from the five CLs were added together and divided by the total number of responses in that Context Area and then multiplied by 100 to obtain a percentage mean.

$$
\begin{aligned}
\text{OMPE-\#CA} \quad &= \sum \text{S(CA)a1} + \sum \text{L(CA)a1} + \sum \text{GA(CA)a1} + \sum \text{ES(CA)a1} + \sum \text{CS(CA)a1} \\
&= \frac{\sum(\sum \text{S(CA)a1} + \sum \text{L(CA)a1} + \sum \text{GA(CA)a1} + \sum \text{ES(CA)a1} + \sum \text{CS(CA)a1})}{\text{TNSCA}} \times 100
\end{aligned}
\tag{5}
$$

OMPE-#CA stands for Overall Multi-level Perception Estimation in a specific Context Area; CA represents Context Area; while S(CA) represents students' perception per concern in Context Area, L(CA) represents lecturers' perception per Context Area, GA(CA) represents graduates/alumni' perception per Context Area, ES(CA) represents employers/supervisors' perception per Context Area, CS(CA) represents clients/service users' perception per Context Area, TNSCA is Total Number of Scores in that Context Area, and 1 = strongly disagree; 2 = disagree; 3 = neutral; 4 = strongly agree; 5 = agree; a = sanitation management content; b = curriculum; c = lecturers; d = learning; e = infrastructure/facilities; f = practicum/internship; g = research project.

### 2.4.2. Interview and Observation

Themes and sub-themes were selected for qualitative analysis of interviews and observations based on the thematic focus of the survey questionnaire, study objectives and response to the survey questions, and then presented as part of respondents' profiles, test of knowledge, teaching and curriculum and respondents' perceptions.

### 3. Results and Discussion

*3.1. Respondents' Profile*

A total of 280 people were surveyed based on varied connections to the EHT programmes of the surveyed college programmes (primary and secondary—see Figure 1)

with 42 percent female and 58 percent male. Evidence shows that there seem to be more males in the profession than females, although enrollment indicates more females, which drops by graduation. About 39 percent were within the 18–27 age range followed by 27–37 (29 percent) and 37–47 (18 percent). Approximately 35 percent of them work with government organisations while over 42 percent graduated from the programmes (see Table 1). About 26 percent of the respondents (polled from students, graduates, employers and lecturers) work as Environmental Health Officers (EHOs), while 20 percent work in the sanitation management sector. In addition, over 63 percent and 55 percent of them desired to work as Environmental Health Officers (EHOs) and specifically as sanitation professionals, respectively.

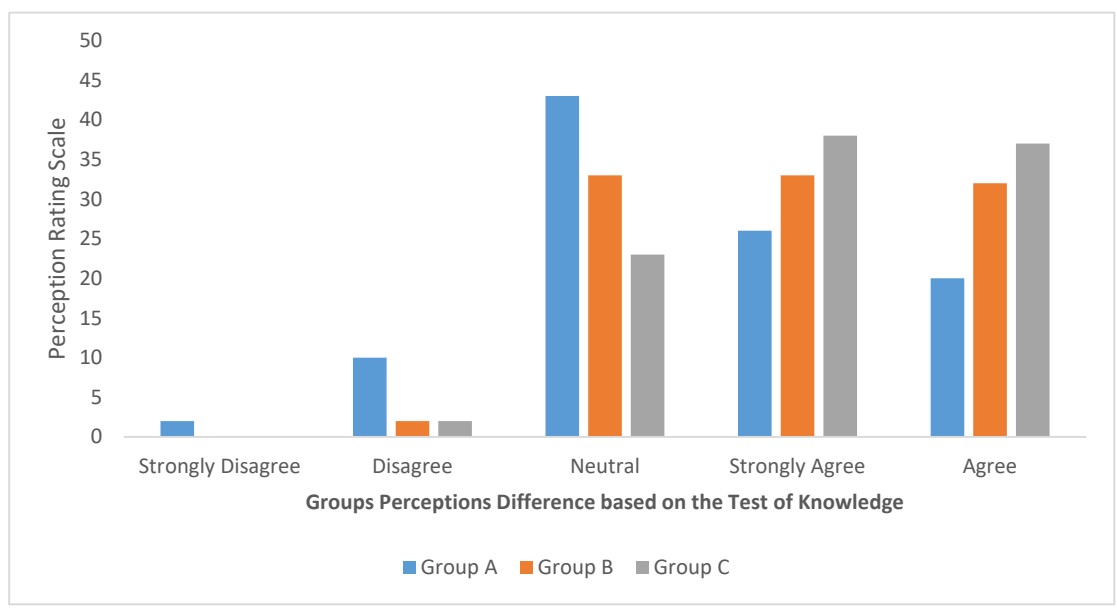

(**a**)

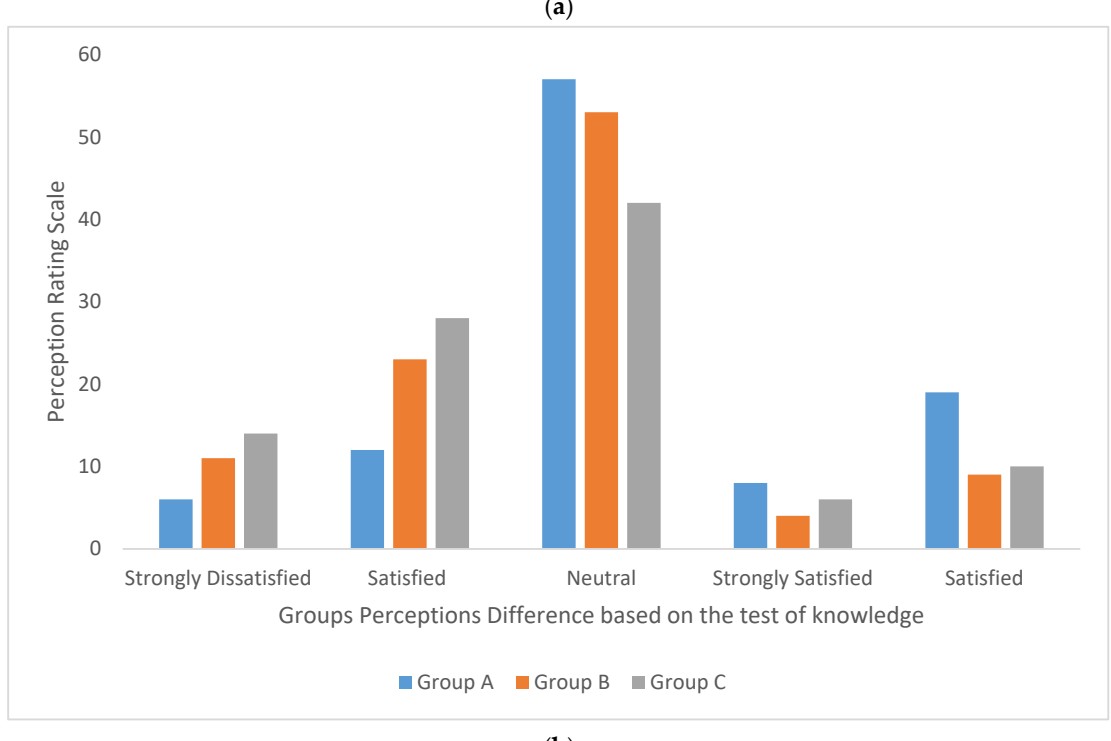

(**b**)

**Figure 1.** *Cont.*

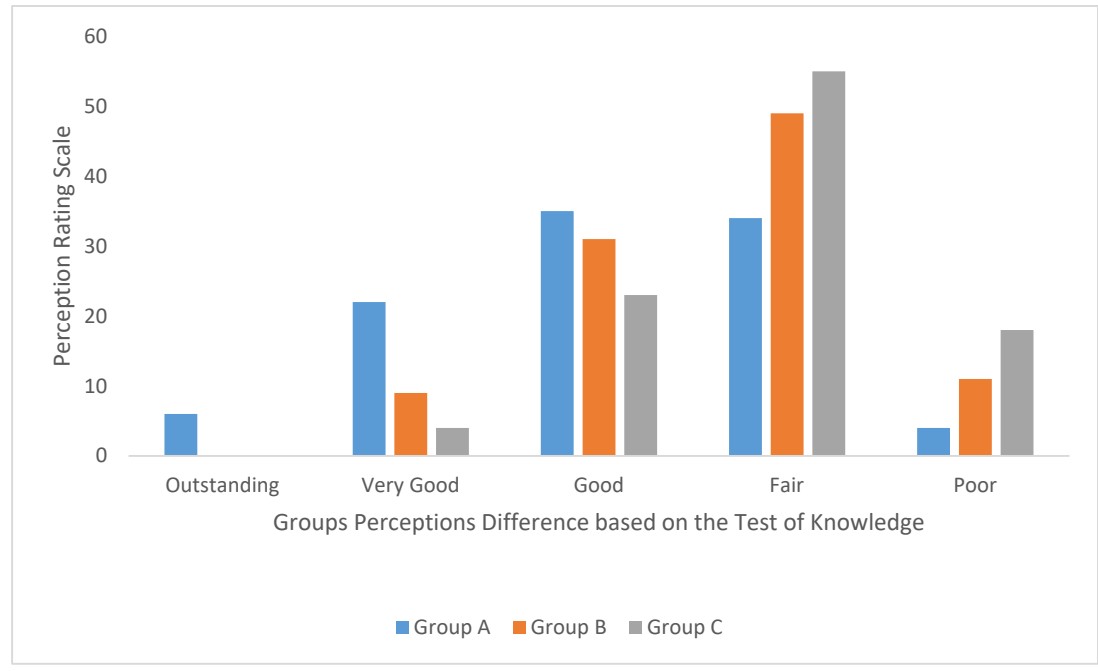

(**c**)

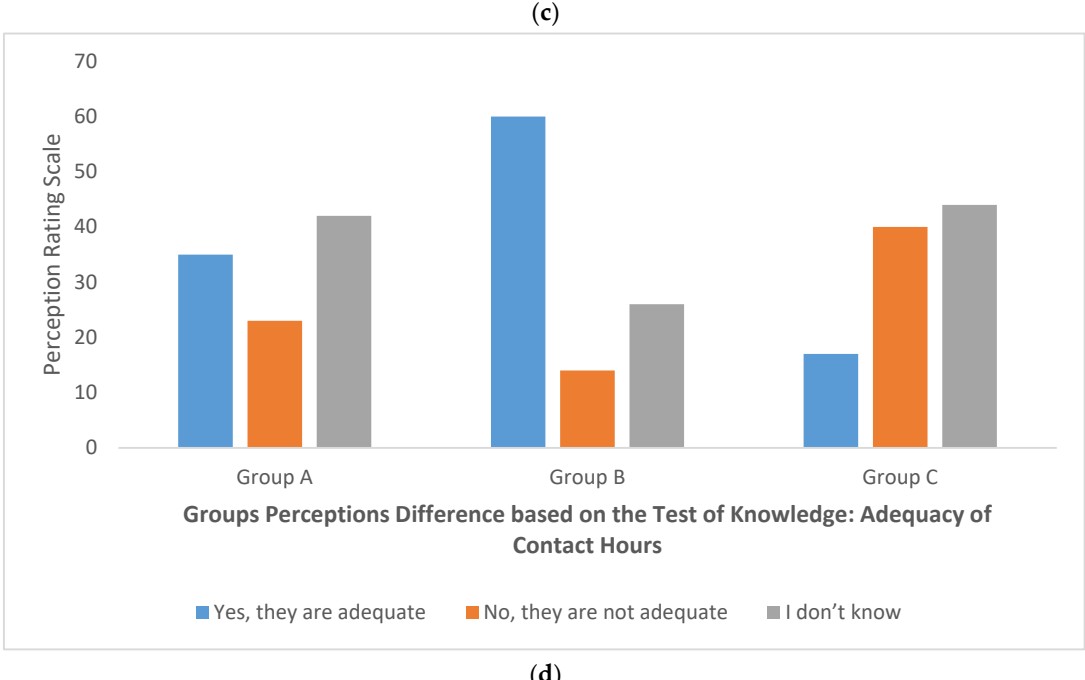

(**d**)

**Figure 1.** *Cont.*

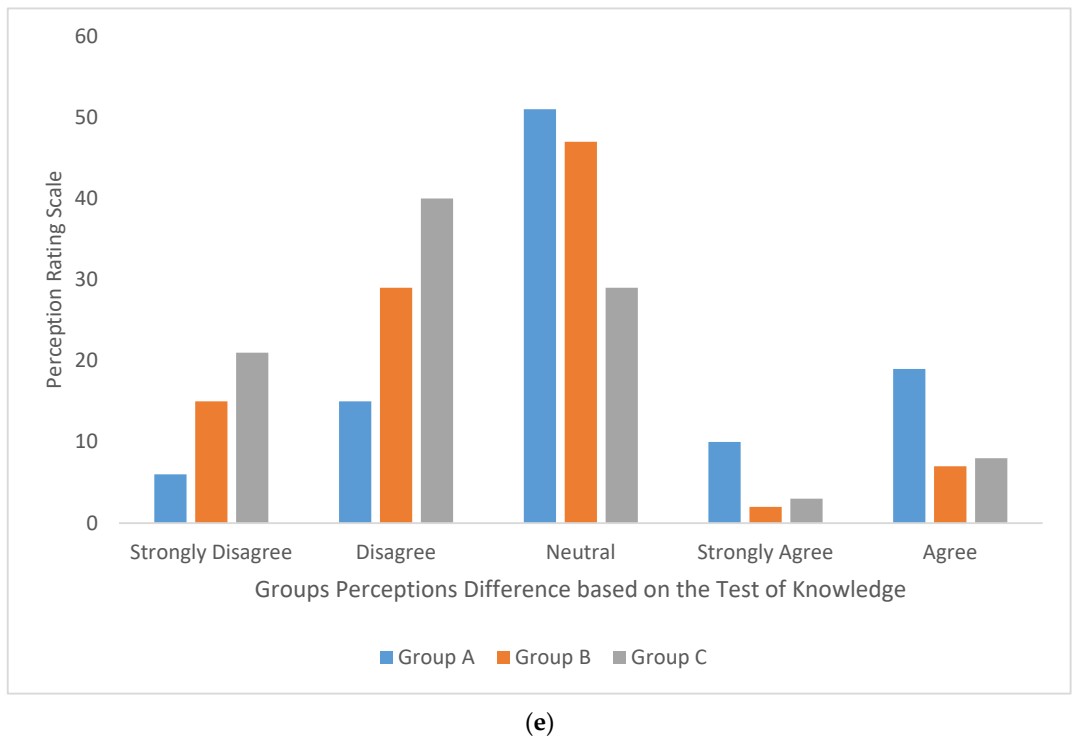

(**e**)

**Figure 1.** (**a**) Group perception differences based on the Test of Knowledge: Sanitation Education Importance in Nigeria. (**b**) Group perception differences based on the Test of Knowledge: Satisfaction with Value and Expectations. (**c**) Group perception differences based on the Test of Knowledge: Rating for Sanitation Management Content. (**d**) Group perception differences based on the Test of Knowledge: Adequacy of Contact Hours. (**e**) Group perception differences based on the Test of Knowledge: Quality of Sanitation Management Content in EHT programme curriculum.

About 70 percent of the students and graduates/alumni were student workers who mostly paid their own way and provided their upkeep through diverse types of employment. In the past, enrolling in the programmes automatically meant being enlisted into the workforce of the state's public service, but this system changed about 20 years ago. Today, most of the current students and past graduates will face a steep employment market, especially in the sanitation/hygiene sector, because it is very poorly defined in Nigeria. This is sad because almost all of the students and graduates surveyed indicated that their top reasons and expectations for enrolling in the EHT programme were to obtain employment (students—100 percent and graduates—78 percent); however, only 6 percent of the graduates have been able to have that expectation fulfilled, as 44 percent indicate that they have not gained much since graduating (Table 1).

However, only about 28 percent of these two category levels of respondents have ever had any form of extra-curricula sanitation management training, while 36 percent are not sure and 30 percent have never had any further training. Some of the trainings they participated in were community sanitation awareness, environmental sanitation awareness, house-to-house sanitary inspection and others during Students Industrial Work Experience Schemes (SIWES). On the other hand, just about 27 percent of lecturers indicated any specialisation in sanitation management with 33 percent indicating no experiences in specifically postgraduate study, research, peer-reviewed research articles or professional qualifications or training in sanitation management, while about 40 percent were not sure.

The employers and clients indicated that EHT graduates from the surveyed programmes operate in assignments and services that include sanitary inspections, solid waste management, sewage disposal, pest control, sanitation awareness, regulations and other such areas in various capacities, mostly as individuals, enterprises, government entities and others like NGOs.

### 3.2. Test of Knowledge Survey Instrument

A simple Test of Knowledge instrument was administered to determine the knowledge levels of the students, graduates and lecturers of the EHT programme on current sanitation management practice. Out of the 223 respondents that took the test, most of them only attempted some 5 to 10 out of 30 questions. Five out of these ten questions were multiple choice that evaluated an understanding of specific registers/terms such as sanitation, sanitation systems, services, facilities and non-sewer sanitation (see Table 2). Almost 80 percent of all respondents showed limited knowledge of these terms and key frameworks, concepts and technologies used in sanitation today. Sanitation was seen as being about making the environment conducive for human habitation and the Test also revealed a limited understanding of the concepts of ecological and sustainable sanitation. Concepts like sanitation behaviour, sanitation service chain, sanitation value chain, citywide inclusive sanitation, community-led total sanitation, and community-led urban sanitation planning were not familiar to most of them. Part Three covered basic knowledge about sanitation resource recovery, global sanitation organisations, sanitation behaviour change intervention frameworks, and Nigeria's ranking in global sanitation coverage. But, over 90 percent of participants did not tackle these questions, and when they attempted the question on ranking, it was clear that they did not know. The test was an eye-opener to the effect that it revealed the shortcomings of the environmental health curriculum as it regards sanitation management for by-products of human digestion [39]. In addition, the participants from category levels 1–3 (students, lecturers and graduates/alumni) expressed different sentiments before and after taking the test. For example, Group B was more critical in the questionnaire survey after taking the test than Group A, who filled out the questionnaire before taking the test.

**Table 2.** Response coverage to questions 1–5 of the Test of Knowledge survey instrument.

| Question 1: Sanitation is (choose one or two options only) | | | | |
|---|---|---|---|---|
| *Question Options* | *Students (122) %* | *Graduates (86) %* | *Lecturers (15) %* | *Sum (223) %* |
| (a) management of public nuisances | 49 | 52 | 40 | 50 |
| (b) act of protecting drinking water from contamination | 15 | 23 | 33 | 21 |
| (c) safe management of excreta and urine with reduced or zero human exposure | 7 | 12 | 13 | 9 |
| (d) hygiene and cleanliness of premises | 54 | 41 | 53 | 49 |
| (e) act of keeping environment free from nuisances and disease-causing organisms | 68 | 64 | 60 | 66 |
| Question 2: Sanitation system is (choose one or two options only) | | | | |
| (a) the technical parts of a toilet facility | 0 | 14 | 13 | 6 |
| (b) the combination of technologies used for sanitation management | 30 | 29 | 40 | 31 |
| (c) the processes involved in sanitation service delivery | 44 | 91 | 80 | 65 |
| (d) the management structure for maintaining sanitation programs | 41 | 45 | 87 | 46 |
| (e) the equipment used for clean-up during sanitation | 61 | 0 | 0 | 33 |
| Question 3: Sanitation services are (choose one or two options only) | | | | |
| (a) services for the safe management of faeces and urine | 15 | 16 | 13 | 15 |
| (b) services for ensuring proper sanitation exercise | 51 | 66 | 27 | 55 |
| (c) services required to keep environments clean and free of pathogenic organisms | 92 | 73 | 73 | 83 |
| (d) services required for sanitation intervention programs | 31 | 28 | 67 | 32 |
| (e) services for maintaining sanitation equipment | 8 | 6 | 7 | 7 |

**Table 2.** *Cont.*

| Question 4: Sanitation facilities include (choose one or two options only) | | | | |
|---|---|---|---|---|
| (a) public toilets and bath spaces | 29 | 26 | 15 | 27 |
| (b) faecal sludge management plant | 20 | 17 | 11 | 18 |
| (c) industrial wastewater treatment plant | 47 | 42 | 14 | 43 |
| (d) none of the above | 0 | 0 | 0 | 0 |
| (e) all of the above | 40 | 55 | 80 | 48 |
| Question 5: Examples of non-sewer sanitation include (choose one or two options only) | | | | |
| (a) VIP latrines and traditional pit latrines | 80 | 76 | 100 | 80 |
| (b) septic tanks and sewage trunk | 21 | 69 | 60 | 42 |
| (c) pipes and wastewater treatment | 7 | 13 | 7 | 9 |
| (d) wells and boreholes | 36 | 14 | 15 | 27 |
| (e) none of the above | 0 | 8 | 0 | 3 |

However, Group C (which comprised some from Group A) were also more critical (sometimes even more so) than previous groups, as they admitted during interviews that the test made them realise that they had a lot more to learn about sanitation than they initially assumed (Figure 1a–e). For example, Figure 1c indicates the differences in perception on rating where Group A participants were alone in rating the sanitation management content of the EHT curriculum as outstanding.

*3.3. Teaching and Curriculum*

An assessment of the central curriculum and specialisations of the teaching staff were reviewed, and the sanitation/hygiene management content of the curriculum was found to be seriously out of date and almost non-existent since the focus is the general concept of environmental/public health, while a minimal number of the teaching staff had specialisations in sanitation management. Table 3 shows the percentage of students who take or have taken these sanitation-related courses, graduates who took them and the lecturers who teach them.

**Table 3.** Sanitation and related courses being taught in the EHT programmes' central curriculum.

| Courses | Students % | Graduates/Alumni % | Lecturers % |
|---|---|---|---|
| Introduction to Environmental Health | 100 | 100 | 20 |
| Water Sanitation | 100 | 100 | 13 |
| Food Hygiene and Inspection | 100 | 100 | 13 |
| Sanitary Inspection of Premises | 100 | 100 | 20 |
| Meat Inspection Hygiene and Sanitation | 100 | 100 | 13 |
| Community Sanitation | 100 | 100 | 13 |
| Sewage and Wastewater Management | 100 | 93 | 20 |
| Pollution Control | 100 | 69 | 20 |
| Entomology and Pest Control | 100 | 100 | 13 |
| Pest Management Method and Control | 52 | 48 | 13 |
| Solid Waste Management | 100 | 100 | 13 |
| Water Quality Management | 100 | 33 | 13 |
| Public Utilities and Environmental Health Issues | 100 | 73 | 13 |
| Occupational Health and Safety | 72 | 76 | 13 |
| Health Promotion and Education | 84 | 100 | 20 |
| Environmental Health Laws, Ethics and Policies | 100 | 100 | 13 |
| Environmental Health Services in Emergencies | 86 | 64 | 13 |
| Environmental Health Monitoring and Impact Assessment | 82 | 62 | 13 |
| Industrial Layout and Landscape Planning/Management | 100 | 69 | 13 |

Meanwhile, about 30 percent of the courses in the 4-year HND (2-year OND (Ordinary National Diploma) and 2-year HND (Higher National Diploma) Bachelor's degree) EHT

programme's central curriculum contain some sanitation-related content with an estimated 300 contact hours (90 percent of this content is in the HND level), but the three-year Environmental Health Technician Certificate programme contains just about 10 percent with approximately 90 contact hours [40,41]. The sanitation/hygiene management content of the central curriculum in both programmes surveyed was found to be inadequate and grossly outdated. This central curriculum is officially approved by the EHCON and WAHEB (the national regulatory bodies) for teaching and practising environmental health management anywhere in the country, especially for those who work with the government at various levels as well as in the private sector. Considering that the focus of the programme is broad general environmental health concerns, sanitation was only given brief attention even though it is expected that graduates from this programme across the country's almost 50 Colleges of Health Technology will be the governments' front-liners in sanitation management. Most of the courses that contain related or maybe even major content are taken by almost all who go through the programme (see Table 3), but as the Test of Knowledge indicates, there is something missing in the training.

There are about 33 lecturers surveyed in this study and an estimated 20 of them teach these sanitation and related courses, 15 of whom participated in our survey (see Table 3). The specialisations of the teaching staff include law, pollution studies, environmental education, environmental management, environmental health science, public health, occupational health education, health promotion and community health, all at postgraduate and PhD levels. However, most of these lecturers do not have direct and core sanitation management training and experience; thus, they teach from the environmental health considerations contained in the curriculum, where sanitation is simply the elimination of public health nuisances.

There are so many courses that could be included in the programme, but because the focus is environmental health management and the many concerns thereof demand attention, it is difficult to give extensive coverage to sanitation management as it deserves. In addition, since most of the lecturers are not aware of current sanitation management practices and knowledge, it limits the skills and knowledge obtained by graduates of these programmes who aspire to be part of the sanitation solution in Nigeria. This, of course, is a challenge for the country's drive towards SDG 6 sanitation targets because the EHOs are the key managers of sanitation. Indications are that training and retraining for EHO graduates and lecturers interested in sanitation management is an urgent requirement.

Nevertheless, 40 percent of category level 1–3 (students, graduates/alumni and lecturers) respondents were of the opinion that the current number of contact hours was adequate (Figure 2), which spanned from 80–140 contact hours. However, 38 percent were not sure if the available hours for teaching and practical training were enough to pass on the skills and knowledge required to function as sanitation professionals while 22 percent perceived that they were insufficient. Some of the reasons include more time should be spent learning about sanitation matters; not enough global coverage in the curriculum; not enough time to provide adequate knowledge; not much to learn about sanitation anyway; and the time allocated is enough and the time allocated is not the problem.

Furthermore, a good number of these respondents were not satisfied with the quality of the sanitation management content (SMC) of the central curriculum of the EHT programme, which is the same curriculum used in all states of the federation and the Federal Capital Territory. Although there was a slim margin in the consideration for the quality of the SMC, 29 percent disagree that it is of high quality, while 28 percent agree; 25 percent disagree that it is of consistent quality, while 24 percent disagree. On the other hand, 43 percent do not consider it to be current and up to date and 33 percent do not believe that it is adequate and appropriate even though 18 and 21 percent, respectively, agree. However, most of the respondents were neutral, probably due to personal bias as most of them study at, teach or have graduated from these programmes (Figure 3).

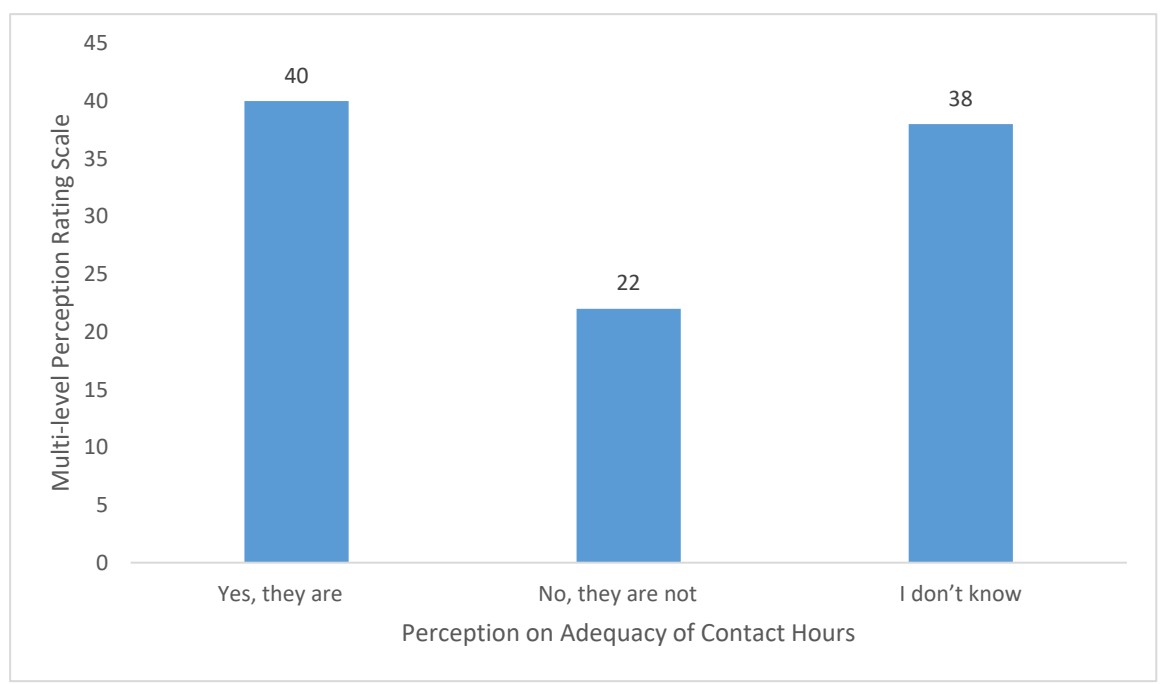

**Figure 2.** Final score of multi-level perception analysis: adequacy of contact hours.

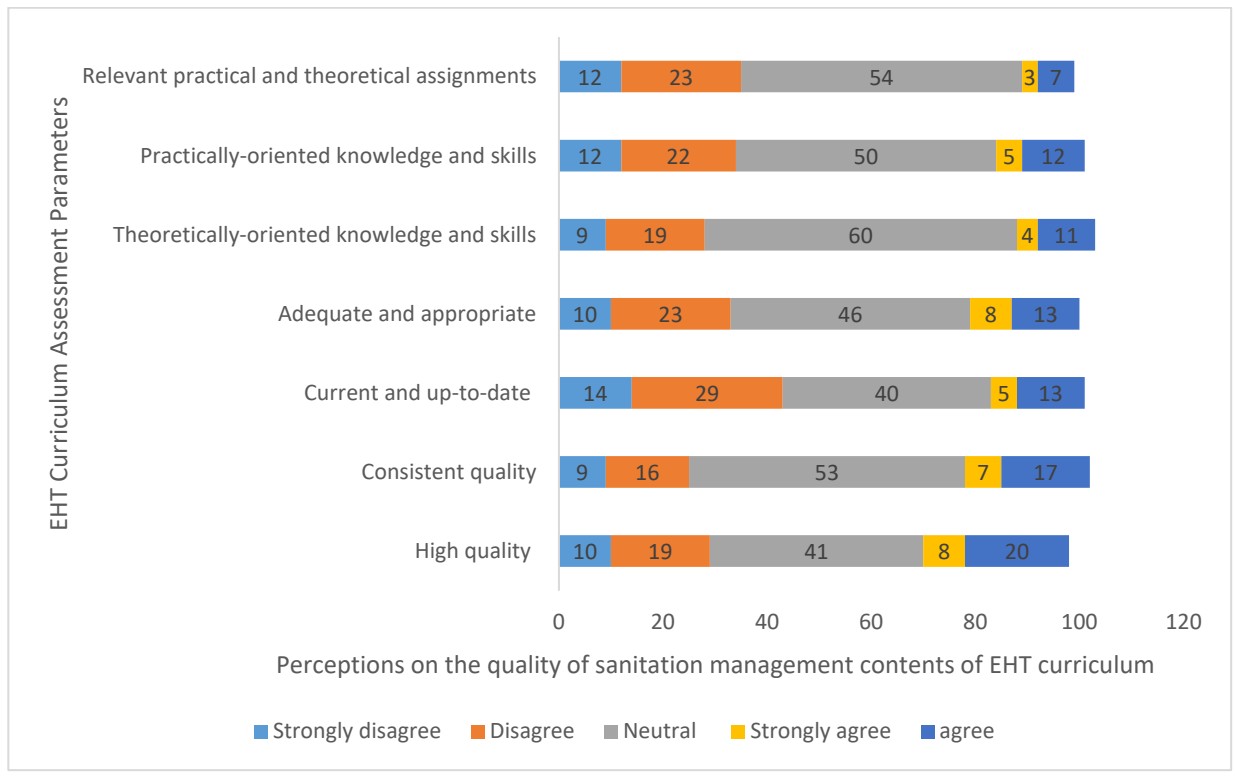

**Figure 3.** Overall perception on the quality of sanitation management contents of EHT curriculum.

*3.4. Respondents' Satisfaction and Rating*

The survey explored the perceptions of all five category levels on their satisfaction with and rating of the sanitation management content and delivery, as well as the quality of graduates from the surveyed programmes.

The respondents were neutral in their satisfaction rating for value added and expectations met, except with employability (48 percent), curriculum content (28 percent), money

spent (32 percent) and practicum experience (30 percent), where there was more discontent than agreeable responses (Figure 4). It is understandable that respondents may not know how to assess the curriculum for sanitation management skills and knowledge transfer as the sanitation sector in Nigeria is not clearly defined and is mostly at the awareness periphery. But, of course, employability is a big issue for these stakeholders because in the past there used to be direct employment with the public service for students enrolled in the programmes, but this has now stopped for over two decades. Most of the graduates are unemployed or work in other sectors while others volunteer in related government entities or NGOs; this is the same plight awaiting existing students. Yet, these stakeholders are a workforce that could be trained as the sanitation army of the country. But, because the curriculum content is not focused on sanitation, research and practicum training are inadequate and the programme cannot deliver the knowledge and skills required to meet the sanitation needs of the country beyond awareness.

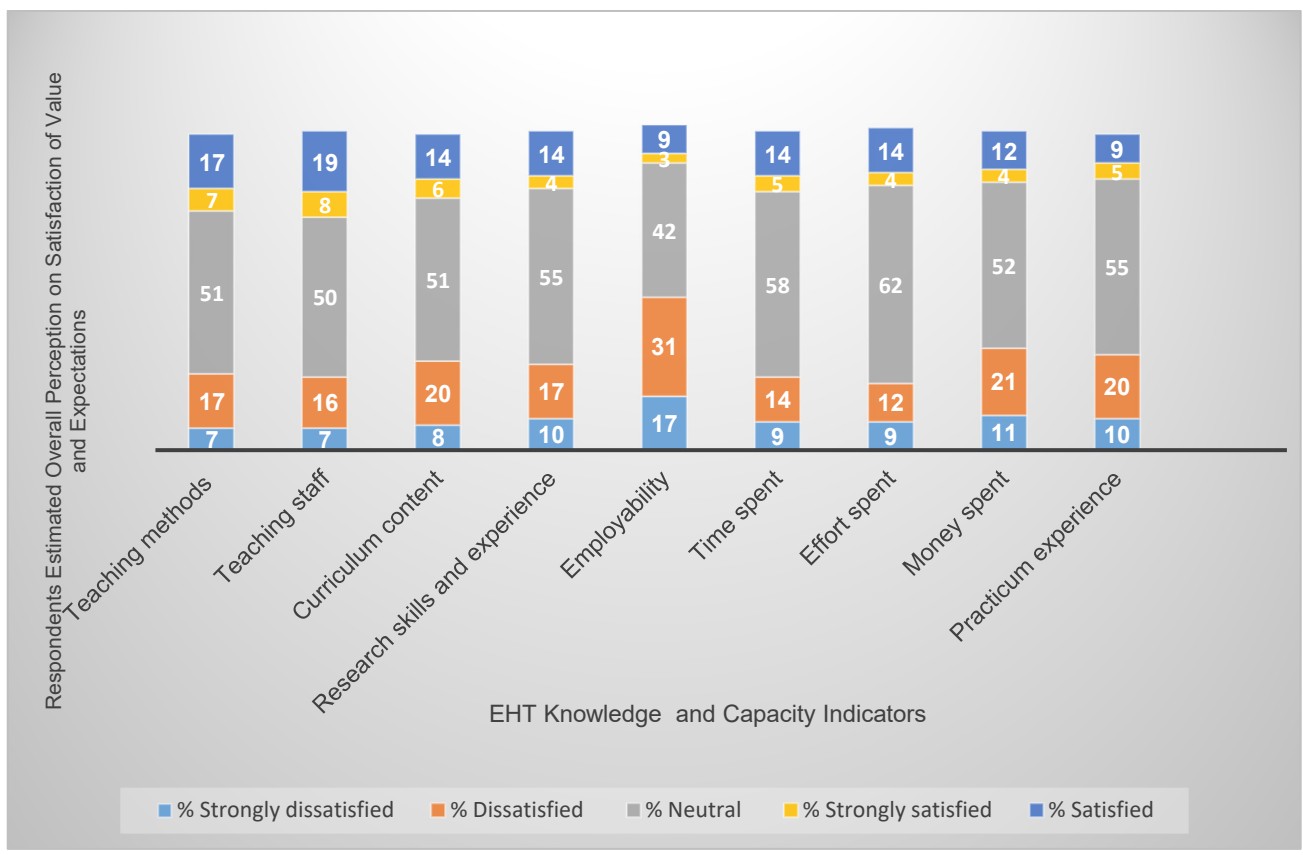

**Figure 4.** Estimated Overall Perception of satisfaction of value and expectations from the EHT programme.

Employers/supervisors and clients/service users also indicated their satisfaction with the skills and knowledge level of EHT graduates from the surveyed programmes. Over 50 percent of these respondents were dissatisfied with the skills and knowledge levels of EHT graduates from the programmes under review. For instance, 54 percent were dissatisfied with their skills and knowledge levels to operate as sanitation professionals, 41 percent did not perceive that they were adequately equipped to operate as government regulators, 61 percent were not satisfied with their knowledge about the science of excreta and urine while 68 percent were dissatisfied with them on their comprehension of sanitation resource recovery and reuse. However, 36 percent and 27 percent, respectively, were satisfied with the EHT graduates' abilities as sanitation/hygiene professionals and government regulators (Figure 5).

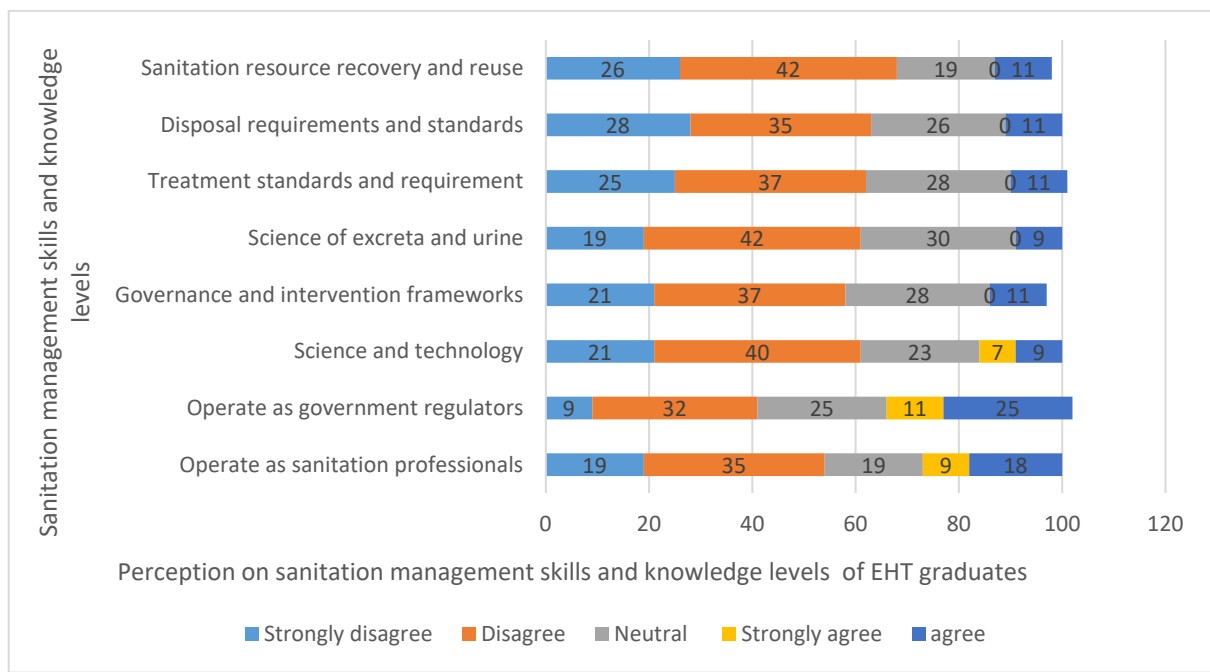

**Figure 5.** Respondents' estimated overall perception of sanitation management skills and knowledge levels of EHT graduates.

Figure 6 shows the overall satisfaction with the skills (competencies) and knowledge levels of these graduates according to employers/supervisors who oversee their assignments in the workplace and the clients/service users to whom they provide sanitation management/regulatory services. This paints a clear picture of the sanitation challenge in Nigeria. If those who are mandated by law to manage sanitation in the country are not adequately trained and equipped to provide the services, technology and governance required, how will the nation make the needed progress?

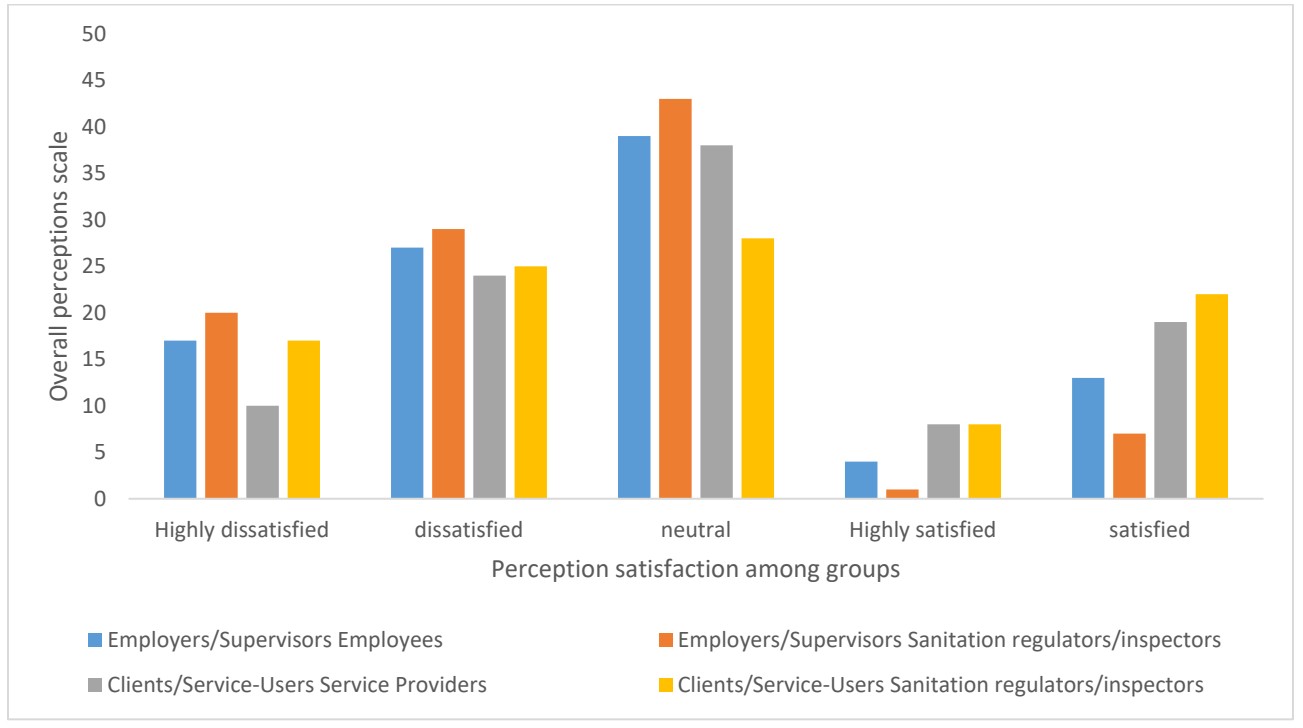

**Figure 6.** Skill and knowledge levels of the environmental health graduates.

The majority of the respondents in all five category levels rated the sanitation management content of the EHT curriculum and programme as fair: (44 percent (students), 42 percent (graduates), 27 percent (lecturers), 53 percent (employers) and 44 percent (clients). A total of 4 percent of students and 7 percent of lecturers rated it as outstanding while 7 percent (students), 8 percent (graduates), 19 percent (employers) and 40 percent (clients) considered it to be of a poor rating (Figure 7). In essence, this is indicative of the fact that the curricula and programmes of environmental health in Nigeria need to be reviewed as it concerns sanitation management.

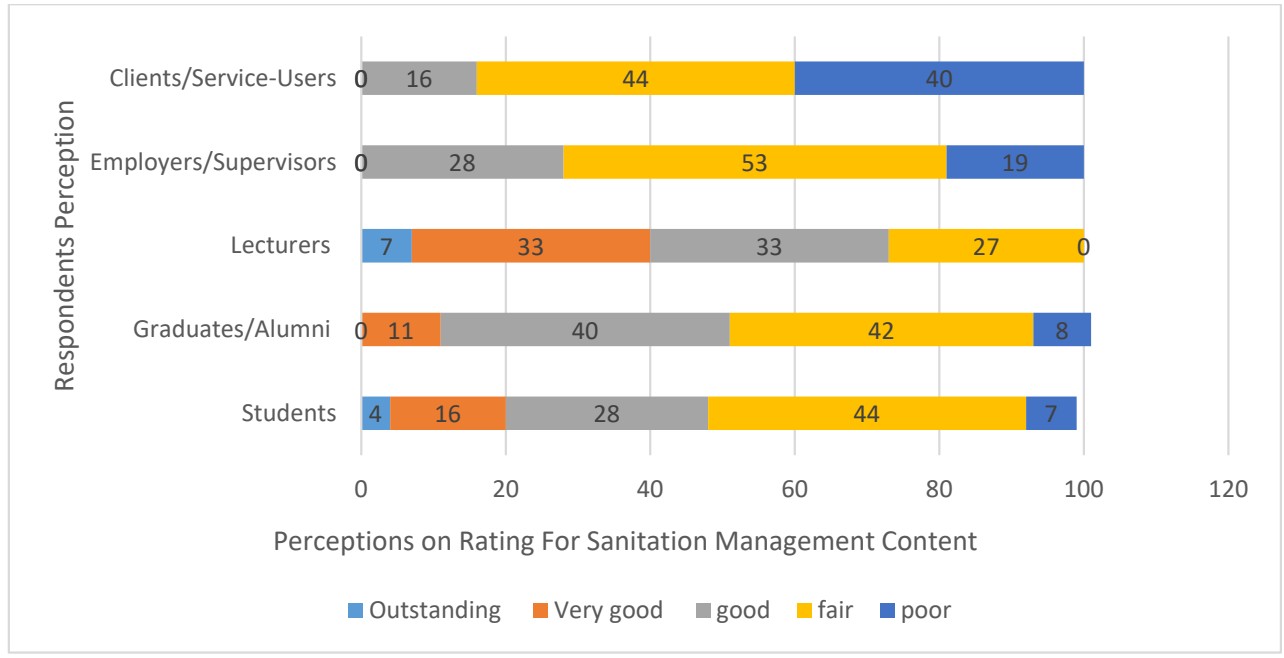

**Figure 7.** Estimated overall perceptions of rating for sanitation management content.

### 3.5. Perception of the Importance of Sanitation Education in Nigeria

As the pace increases towards 2030, indications show a paradigm shift that views sanitation matter as a resource, which when paired with inclusive and collaborative management will provide better results than the reactive exposure/reduction measures [42]. With the increasing evidence of slow progress towards ensuring access to safe sanitation solutions, especially in developing countries like Nigeria, it becomes expedient to find new ways to achieve effective and viable sanitation management that works. Solid waste aside, by-products of human digestion (e.g., faeces and urine) and materials from other directly related human hygiene activities should be major concerns for sanitation managers [43], as the impact could have a rippling effect on all other sustainable development goals (SDG) and then erode the gains in other sectors such as health, education, livelihood, equality and sustainable urban development. It is crucial to develop a crop of professionals that comprehend the whole spectrum of sanitation/hygiene challenges with innovative, systemic and integrated management systems that work. Thus, the country needs to make sanitation/hygiene management education a priority and ensure appropriate, sufficient, and up-to-date knowledge and skill transfer [44]. It is, therefore, no surprise that all of the survey participants agreed that high and consistent quality sanitation management education is a necessity and priority and that higher education/TVET institutions should provide such programmes (Figure 8) to boost the quality and population of sanitation professionals in the country.

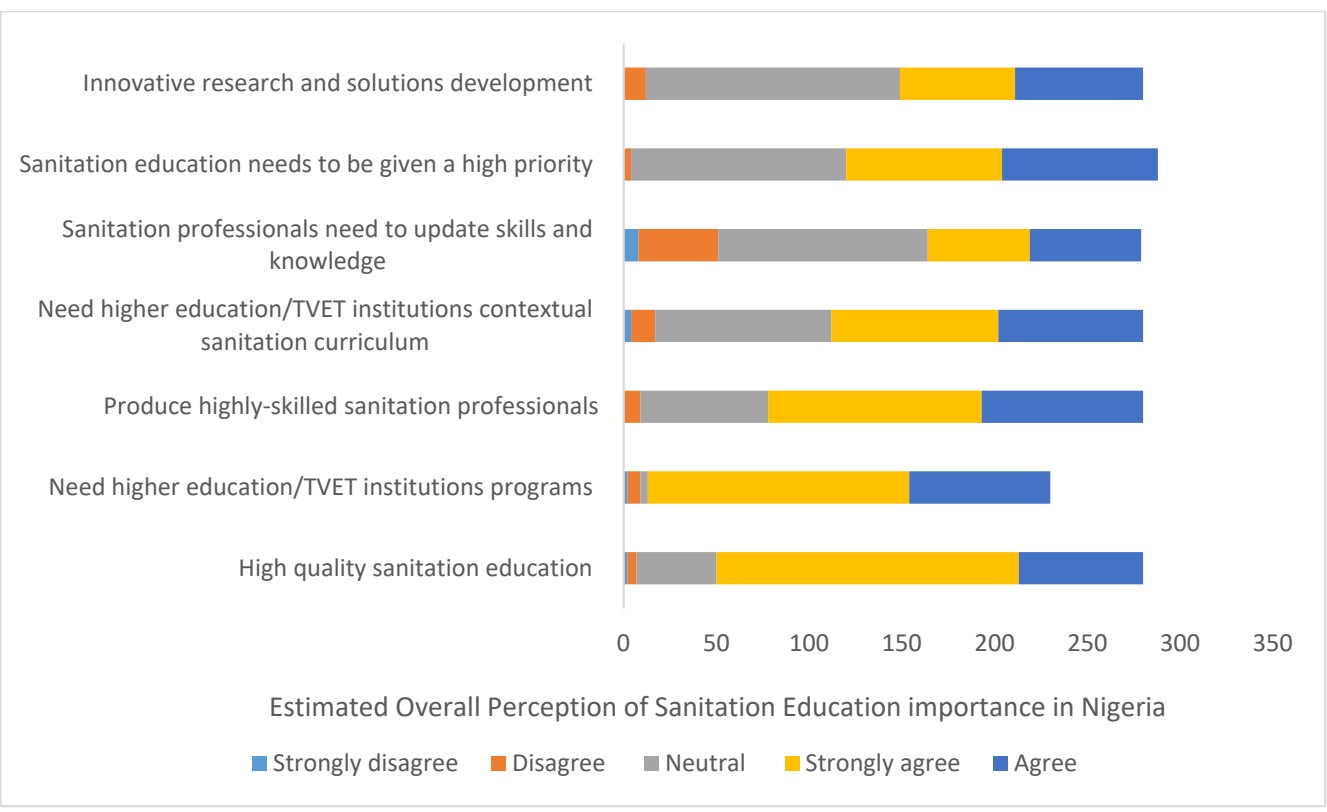

**Figure 8.** Estimated overall perception of sanitation education importance in Nigeria.

*3.6. Multi-Level Perception Score on Sanitation Management Content of the EHT Programme Curriculum*

In summary, Figure 9a–c show the final perception score on the different key aspects of the study and indications are that the respondents agree (sometimes strongly) that sanitation education is important, but employers and clients are dissatisfied with the skills and knowledge levels of EHT graduates on sanitation management. And, although most of them were neutral on the quality of the sanitation management content of the EHT programme/curriculum and in their satisfaction with the value-added and fulfilled expectations quotient of the programmes, they generally rated them as fair. Across the five category levels, Figure 8 shows that overall perception was mostly neutral (53 percent) for added-value and fulfilled expectations by the EHT programmes and curricula because most of the respondents could not determine outcomes as they did not really know what was expected. However, some were strongly dissatisfied (10 percent) and dissatisfied (19 percent) while others were strongly satisfied (5 percent) and satisfied (14 percent). Apparently, more people were not satisfied with the programme delivery and curricula. But, employers/supervisors and clients/service users in Figure 8 were mostly strongly dissatisfied (21 percent) and dissatisfied [38] with the skills and competencies of graduates of the surveyed EHT programmes, even though 13 percent indicated satisfaction while only 3 percent were strongly satisfied and a quarter of them were neutral. Consequently, the challenge of sanitation management is evident in the inability of those who were trained to manage the system to deliver competent and effective services and tasks.

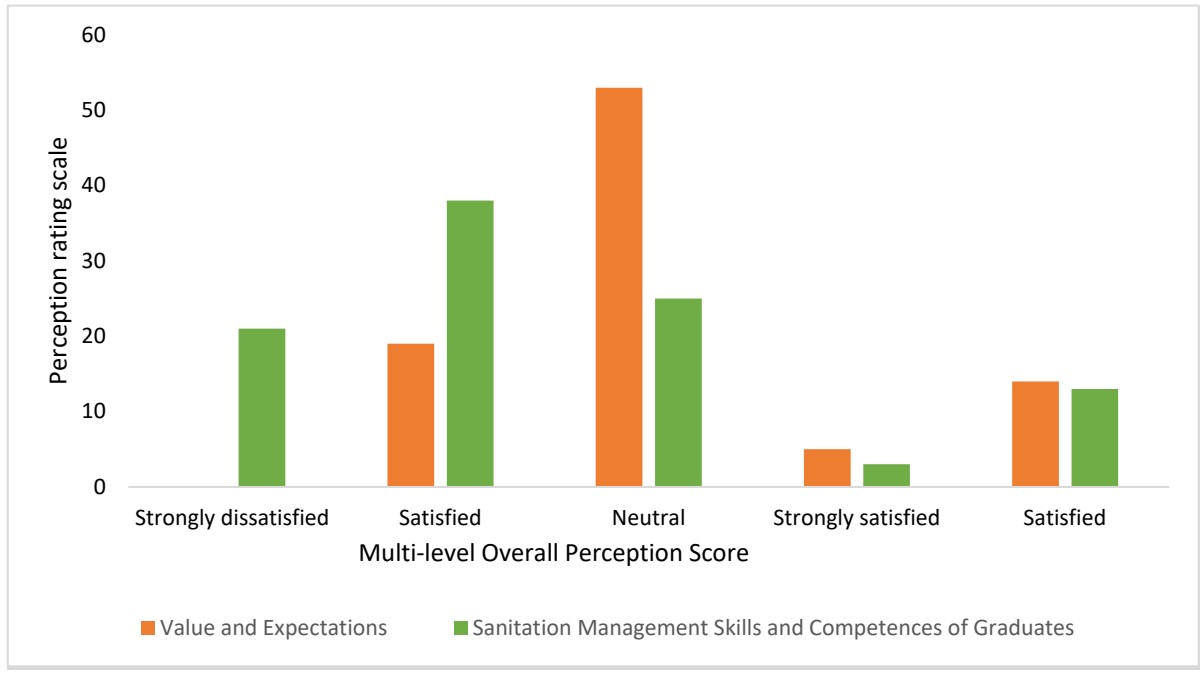

(**a**)

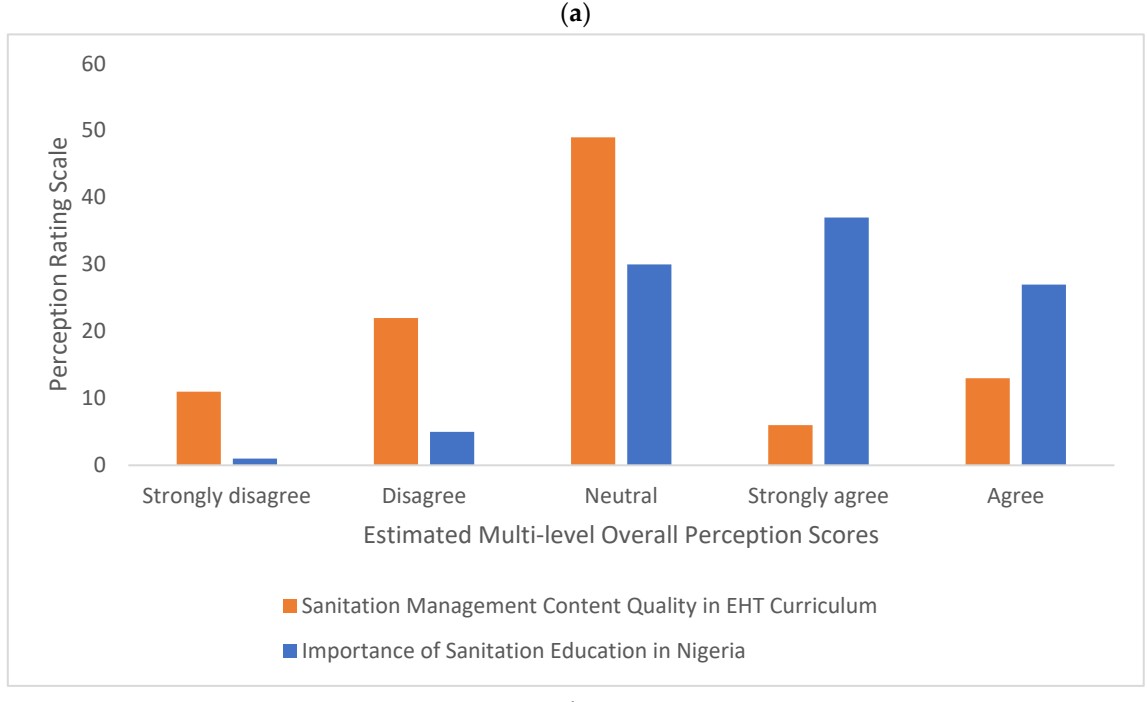

(**b**)

**Figure 9.** *Cont.*

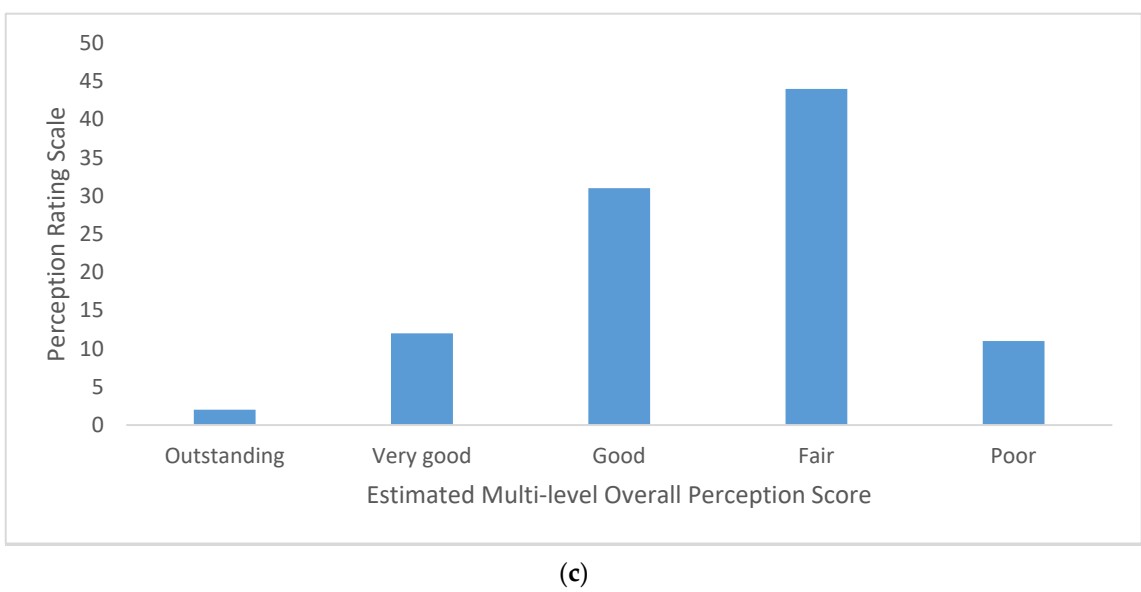

(**c**)

**Figure 9.** (**a**) Respondents' estimated Multi-level Overall Perception score: value and expectations/sanitation management skills and competencies of graduates. (**b**) Respondents' Estimated Multi-level Overall Perception scores: sanitation management content in EHT curriculum, and importance of sanitation education in Nigeria. (**c**) Respondents' Estimated Multi-level Overall Perception score: rating of sanitation management content in EHT curricula and programme delivery.

Perhaps this challenge could be emphasised in the fact that overall scores from students, graduates/alumni and lecturers show that more of the respondents do not agree that the sanitation management content of the EHT programme curricula is of quality standards (strongly disagree 11 percent and disagree 22 percent), while just 6 percent strongly agree and 13 percent agree. However, a 49 percent score for neutral is indicative of the fact that most of these respondents were hesitant to be critical about their current institution or alma mater. Also, it should be noted that results for Group C in all these category levels were not computed for the overall results and most of them were more critical after being unable to complete the ToK. Nevertheless, most respondents in the five category levels were in agreement that sanitation management education is important in Nigeria's fight to provide safely managed sanitation services and products. A 37 percent score indicated strongly agreed while 27 percent agreed; however, 1 percent strongly disagreed and 5 percent disagreed, with 30 percent remaining neutral. The case of poor employability for graduates in the past 20 years and more could be responsible for the neutral consideration.

In the end, the multi-level overall perception analysis revealed that 44 percent of all respondents rated the sanitation management contents in the EHT programme and the programme delivery as fair while 31 percent rated them as good. A 2 percent score went to outstanding and it all came from the lecturers. Meanwhile, 12 percent score went to very good and poor obtained 11 percent (Figure 9b,c). It is evident that the survey participants generally perceive that the programmes and curricula need some major improvement as it concerns sanitation management even though it does cover some other key aspects.

### 3.7. Interview and Observations

The majority of the interview respondents agreed that the sanitation management content of the curriculum was inadequate, inappropriate and outdated and would not be enough to tackle the challenges that the country faces. Most employers and clients surveyed also agree that many of the EHT graduates from the central EHT programmes are not duly equipped to carry out effective sanitation management. Generally, they rated the EHT programme's sanitation management content as fair with a lot of room for improvement. In conclusion, it was agreed that sanitation education is of great importance to the 2030 vision of the country. Specifically, issues of not having adequately knowledgeable and competent

personnel teaching specialised courses in sanitation management were highlighted, as well as the need for the lecturers, and indeed field workers, to upgrade their information and knowledge so they could teach better and work more effectively. The challenge of grossly inadequate facilities in the institutions of learning and a lack of public respect for sanitation professionals was also a concern. One recurrent issue was how the lack of employment opportunities has made it difficult to even practice or desire to update and/or upgrade skills and knowledge. The interview respondents who partook in the ToK were shocked at how insufficient their knowledge and skills in sanitation management were even after 4 years of academic experience in environmental health and many years of working in the sector, and they were really keen on their aspirations to acquire more information, knowledge and skills to equip them as sanitation management professionals.

## 4. Lessons Learned and Recommendations

It is clear that the programme designed for those professionals (environmental health) mandated by law to manage sanitation and hygiene in the country does not transfer the needed knowledge and skills. This is understandably so because environmental health is much bigger than sanitation and can only view sanitation from the perspectives that are relevant to public health. Recent concerns have shown that sanitation management is bigger than it was traditionally viewed and has far-reaching tentacles into other disciplines besides health and water. This indicates that to create a pool of sanitation, professionals will require a new way of thinking about sanitation education and training for those who will work to ensure safely managed processes across the service and value chain in the country. The EHT programme curriculum could be upgraded with specialisations in sanitation/hygiene; however, this may still not be enough. There is an urgent need to train the sanitation workforce in the region at all levels on the changing trends and new approaches to delivering regenerative and circular services to an increasing population and varied contexts in the race towards 2030. It is expedient that researchers begin to look into this area.

One of the key lessons learned from this survey is the fact that the sanitation/hygiene management content of the environmental health curricula needs to be updated and upgraded and the skills, knowledge and competencies of current sanitation managers (especially in government) need to be updated and upgraded as well. The sector needs to be clearly defined and enabled to deal with the issue of employment opportunities as well as service and product availability. Sanitation management education is crucial in the race to 2030 and cannot be left to the environmental health education programmes alone.

The challenges of sanitation/hygiene management seem to be taking on new dimensions. Even as efforts intensify to change behaviour that increases human exposure and offers technological solutions, there is still a wide margin between desired results and reality [45–48]. In addition, other sustainable development goals and aspirations of the country could be more achievable when sanitation is properly positioned in the planning process and delivered in a holistic manner. This is essential because sanitation involves key activities of the urban population that produce bodily and other matter, which, although it has proven to be hazardous to public health and environmental quality [49], also has huge resource potential (when treated and rerouted) for the sustainability of cities and their populace [39,48,50,51]. This will require the design and delivery of context-specific education and training. One offshoot of this study that needs to be investigated is why more females enrol at the beginning of these programmes, but only very few graduate at the end, and what can be done about it.

Taking all these into consideration, a deliberate and structured design to make sanitation and hygiene management HE a focused discipline area should be a priority at all levels of government. Meeting the SDG 6 targets on SH by 2030 and sustaining the gains will require a competent and efficient workforce, as well as versatile and viable enterprises that provide innovative and contextual solutions and services [39]. It has been observed by scholars that SH needs to be addressed on its own platform because managing and

providing services and products to users are unique and major enough to even create a sanitation economy [39,52,53]. Additionally, the process of sanitation service provision is linked to technological, institutional, psycho-social, socioeconomic, ecological, geographical, cultural, governance, business and other related disciplines. It should, therefore, be treated as a central theme of its own in order not to tackle its peculiar and contextual demands. The development of innovative solutions such as technology that is designed to align with nature, provide contextual products and services and specifically for the population at the base of the pyramid (BoP) will require a system that develops and equips a workforce with the specific skills and knowledge to deliver acceptable solutions and appropriate infrastructure. Short and basic training conducted for and by civil societies (NGOs/CBOs) and academic programmes that focus on intervention cannot deliver these kinds of capacities.

Furthermore, just making learning available to grow a crop of sanitation/hygiene management professionals and practitioners will be antithetical to progress if they cannot obtain employment because there is no demand for their skills and competencies in a system that has no clearly defined sanitation/hygiene sector or sub-sector as the case may be. In essence, institutional (policy, legislation, regulations, standards, etc.) and financial support [54] as well as enabling enterprise development will fundamentally improve national capacity and knowledge base. It has been suggested that sustainable transitions could arise from the discovery and expression of new ideas, narratives, practices, governance instruments and solutions that could change institutions, infrastructure, technology, behaviours, services and economies, among others [55–57].

Meanwhile, environmental health professionals are currently the government officers with the responsibility of managing sanitation in urban/rural areas and they are the regulators, planners and guide policy makers' decisions. These officers are generally trained through the environmental health education system of the EHCON and WAHEB education/curricula and professional development training and licensing of the EHCON. However, the sanitation management content of the EHCON curricula is grossly outdated and inadequate because, understandably, sanitation is not the focus of the discipline and there are several other concerns to cover. This indicates that sanitation management is bigger than the space and time allocated to it in the environmental health programmes even though these programmes provide a strong foundation for sanitation management. It is, therefore, evident that environmental health professionals serving as sanitation management officers need to upgrade their skills and knowledge to be effective and competent in ensuring safely managed sanitation practices in the country at all levels. And so, this paper proposes that sanitation/hygiene management be treated as a discipline on its own and a field of specialisation for environmental health professionals.

## 5. Recommendation: Transformative Pathway for Sanitation and Hygiene Management Education in Nigeria

This paper recommends a national sanitation/hygiene management higher education (NSMHEd) pathway (Figure 10) to guide the provision of integrated sanitation/hygiene management education at HEIs, technical and vocational education training institutions (TVET) and professional/practitioners levels to build effective, efficient, competent and workforce for the country's drive towards the 2030 sanitation targets of SDG 6. It will involve a systemic approach that is needs, competency and inter-profession based with a rich environment for research and development as well as practical linkages with industry and governance. The needs will be addressed at local, state, national, regional and international considerations and interconnections. These needs will require the development and provision of products and services that will draw upon broad and specific competencies demanded and expected by employers and consumers alike. The way it all relates to the vision to end open defecation and ensure safely managed sanitation among others guides the facilitation of a comprehensive and integrated sanitation management education (SMEd) development that

could guarantee competencies for sanitation management professionals to deliver related products and services [58–60].

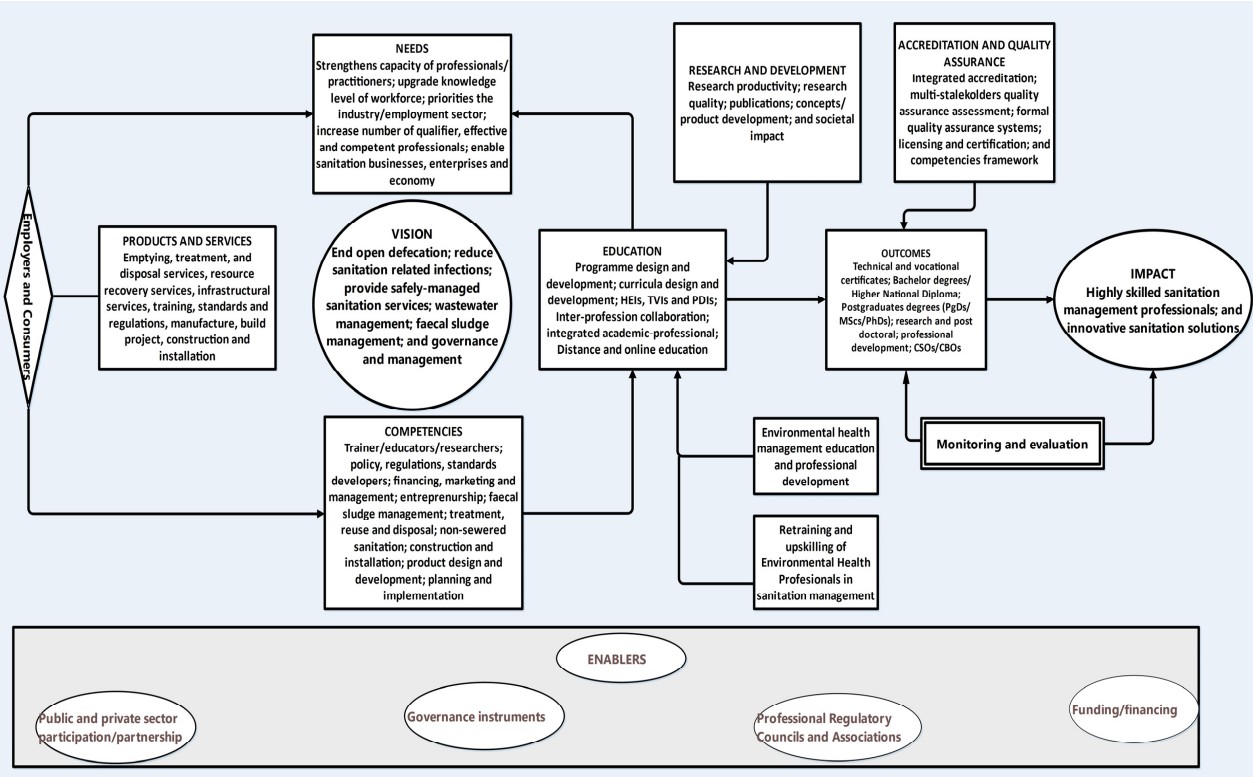

**Figure 10.** National sanitation management higher education transformation pathway.

The aim of the pathway proposed from this study is to serve as a guide to structure and design an academic and professional route to increase the number of competencies in SH management in the country so as to cover up the lapses in capacity and knowledge highlighted in the National WASH Action Plan and other studies and also increase the number of competent and effective professionals. This is the ultimate impact desired from the vision with outcomes whereby HEIs, technical and vocational institutions (TVIs) and professional development institutions (PDIs) will be able to present standard national and global qualifications at various levels. A SMEd that recognises the links between education/training and the SH needs of the country as well as its global effects should be able to equip graduates with skills and knowledge to be competitive and effective in their home and global sectors wherein they can exhibit competence and the ability to transfer knowledge. Essentially, there should be a bridge between competence and skilled expertise for industry and governance in the SMEd. This could then breed high-quality sanitation managers capable of raising the standards of sanitation accessibility and service delivery in the country [58,60].

Thus, as reflected in Figure 9a–c, after determining the needs and scale at different levels of government administration and ecological/geographical conditions, the products, infrastructure and services required per level would then determine the competence demand for each need, which will, in turn, be able to guide programme structure and design, curriculum content and delivery methods to enhance, maintain and sustain the availability of required competencies. In addition, upgrading the environmental health management education structure that provides qualifications for the governments' sanitation managers as well as the professional development training designs will not only improve the quality of their output, but also produce trainers and educators for the SMEd programmes. Also, these government officers could be retrained and upskilled to upgrade and update their professional competence and knowledge so they perform their tasks better.

The NSMHED Pathway programme design, curricula development, HEIs, TVIs and PDIs (public, private, associations, faith based) buy-in, inter-profession collaboration, integrated academic–professional programmes (e.g., professional graduate diplomas) and distance and online structured programmes (particularly in the face of the effects of COVID-19 and social distancing). The interactions between the design and development of products and infrastructure as well as the provision of services with appropriate competence and knowledge displayed could be determined and assessed by prescribed competency frameworks or systems for professionals and practitioners while licensing based on this competency framework among other criteria could be used to secure quality and eligibility in the workforce. Meanwhile, research and development should provide outcomes that make use of competencies while updating them, and then support product/infrastructure and service development and delivery, as well as governance and management. This will contribute greatly to impacts in access expansion, service delivery and innovative solutions. It then means that programme accreditation will address the overall ability to provide graduates with new skills, broad knowledge and a wide range of competencies favourable in a cross-disciplinary sector like SH management. Therefore, it is proposed that accreditation entities will control programme delivery and define the quality, service and route of delivery required to ensure desired competencies, and also provide a basis for positive competition between education and training providers (e.g., benchmarking). Regulatory agencies like the National Universities Commission (NUC), National Board of Technical Education (NBTE) and Environmental Health Council of Nigeria (EHCON), and other related regulatory councils could work together to accredit these SMEd programmes or, better still, an integrated accreditation entity could be established.

On the other hand, Quality Assurance is required on a regular basis after accreditation (e.g., 4–5 year intervals) to ensure that these programmes are still maintaining excellent standards so that employers, service providers, product/service users and graduates will be satisfied with the competencies acquired and/or exhibited. It will be able to guarantee relevance to local and global relevance as the dynamics change and ensure sustainability. This will require a formal QA system or framework for the SMEd programmes and degree qualifications, prescribed standards and quality criteria, governing regulations and professional/industry guidelines/procedures. Furthermore, professional associations, peer (institution's students and academic) community, alumni community, industry, civil society and government ministries and agencies and others will make up assessment target group protocol to be used for QA certifications. Considering that quality is multidimensional, Master's and Phd accreditations and QA should be based on research productivity and quality, publications, concept/product development and societal impact.

The proposed pathway recognises that none of these aspirations can be achieved without the right environment with existing and appropriate factors that enable strong deliverables for a SMEd. Thus, certain enablers are considered:

(i) Public and private sector participation and partnerships whereby public and private heis, tvis and pdis are provided with appropriate resources and incentives to offer sanitation management programmes at whatever levels (certificate, ordinary diploma, Bachelor's/higher diploma, Master's, PhD, research and professional) they can. Also, the industry should be encouraged to partner with education/training institutions to deliver SMED;

(ii) Governance instruments, which include policies, laws, regulations, standards, guidelines and procedures, should provide a backdrop and anchor that creates a foundation for the development of a sanitation sector and economy for graduates of SMED programmes and sanitation management professionals to operate in and fit into;

(iii) Enterprise development whereby resources and incentives are made available for investors and entrepreneurs to establish and grow sanitation businesses and enterprises which will then provide employment opportunities to the graduates from SMED programmes;

(iv) Professional regulatory councils and associations linked to sanitation management could find a way to work together while the formation of specific councils and associations for sanitation management professionals could be considered;

(v) Funding and financing for SMED programmes could be provided through government grants for programme design and development, government support for salaries of highly skilled educators and quality facilities, subsidised tuition fees, especially at postgraduate levels, scholarship and fellowship grants for Master's and Ph.D. candidates and research projects. In addition, funding from industry, government ministries/agencies, national/global multilateral agencies, intervention organisations and individual/organisational foundations could fund infrastructure (e.g., learning rooms, offices, etc.), facilities (e.g., libraries, laboratories, ICT rooms, online learning platforms, etc.), internship training and academic exchange programmes.

## 6. Conclusions

It is recommended that a NSMHEd Pathway in Nigeria should reflect a wide range of competencies required to build a career in sanitation management and to improve the sanitation landscape and architecture of the country and beyond. The aim should be to create a locally trained, competent and sustainable workforce who are capable and flexible with a dedication to safely managed sanitation to meet the increasing demands in the country and globally. The pathway presented here is designed with the Nigerian context in mind to be socially acceptable, evidence-based and competence proven [60] and allows for cross-disciplinary education because SH cuts across all academic disciplines. There are, however, limitations in the shortage of expert and knowledgeable people to teach sanitation management at the higher and professional education levels since it is a new area. It could be possible to explore the option of borrowing from related disciplines and existing programmes with related content and import for specialised areas until enough educators are qualified from the SMEd to fill the gaps. However, the challenge of financing shortfalls could affect infrastructure and facility development as well as the employment of quality academic personnel, which is a common concern for Nigerian institutions.

**Author Contributions:** Conceptualization, P.E.C. and M.A.P.-C.; methodology, P.E.C. and M.A.P.-C.; validation, D.B. and P.E.C.; formal analysis, P.E.C. and M.A.P.-C.; data collection, P.E.C., M.A.P.-C. and I.U.R.; writing—review and editing, P.E.C., M.A.P.-C. and D.B.; supervision, D.B. All authors have read and agreed to the published version of the manuscript.

**Funding:** This reserch receved no external funding.

**Institutional Review Board Statement:** Not applicable.

**Informed Consent Statement:** Informed consent was obtained from all subjects involved in the study.

**Data Availability Statement:** Data is contained within the article.

**Conflicts of Interest:** The authors declare no conflict of interest.

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
