# Peer review of "Multi-Level Perceptions on Higher Education Development for Sanitation and Hygiene Management in Nigeria"

_2673-947X, doi:10.3390/hygiene3040035_

Round 1

Reviewer 1 Report

Comments and Suggestions for Authors

On account of the manuscript HYGIENE-2652955, entitled “Multi-level Perceptions on Higher Education Development for Sanitation and Hygiene Management in Nigeria” by Peter Emmanuel Cookey et al., the authors investigated the quality of the sanitation/hygiene management content of Environmental Health programmes in Nigeria through higher education institutions. The topic is important to better understanding of the current situation of sanitation and hygiene management in Nigeria, and to conduct human health management as well. After careful consideration, I feel that this manuscript is to be published after improvement of some major shortcomings. Details of my comments are as follows:

1) The manuscript is not well written and organized, too long and not well structured, and the readers were difficult to follow up precisely the results and discussion that the authors aimed at the present report. Somewhat redundant statements were observed throughout the present manuscript. The authors are strongly encouraged to make clear and concise descriptions, and to consider including as Supplemental materials those parts of the main text that are incidental to the main thesis.

2) In relation to 1), Introduction is not well structured. The authors don’t necessarily mention general issues in detail, but are better to show information in a summarized way with focusing on the main issues related to the originality of this study. The authors are also better to arrange the description in a much more summarized one.

3) Due to the wide range of topics covered in this paper, the authors are better to have a summary statement of the structure of the present investigation.

4) The format of this paper is not MDPI. In addition, the placement of Figures and Text is awkward, the number of pages is far too high, and the manuscript itself is difficult for the reader to read. The authors are strongly encouraged to format the paper so that it is properly formatted as an Article.

Author Response

Reviewer 1 Comments

The manuscript is not well written and organized, too long and not well structured, and the readers were difficult to follow up precisely the results and discussion that the authors aimed at the present report. Somewhat redundant statements were observed throughout the present manuscript. The authors are strongly encouraged to make clear and concise descriptions, and to consider including as Supplemental materials those parts of the main text that are incidental to the main thesis.

Reply to the Reviewer 1

The comments considered; introduction reviewed and reduced; results and discussion reviewed and some results moved to appendix

Reviewer 1 Comments

In relation to 1), Introduction is not well structured. The authors don’t necessarily mention general issues in detail, but are better to show information in a summarized way with focusing on the main issues related to the originality of this study. The authors are also better to arrange the description in a much more summarized one.

Reply to the Reviewer 1

Introduction reviewed and reduced

Reviewer 1 Comments

Due to the wide range of topics covered in this paper, the authors are better to have a summary statement of the structure of the present investigation

Reply to the Reviewer 1

Summary statement of structure included in the introduction

Reviewer 1 Comments

The format of this paper is not MDPI. In addition, the placement of Figures and Text is awkward, the number of pages is far too high, and the manuscript itself is difficult for the reader to read. The authors are strongly encouraged to format the paper so that it is properly formatted as an Article.

Reply to the Reviewer 1

Figures and text awkwardness addressed. The paper was formatted by the word template provided by MDPI – Hygiene Journal

Reviewer 2 Report

Comments and Suggestions for Authors

Introduction. Shorten the paragraph on the role of higher education as an element of social development. Too general statements. What matters is, how academic sanitation and  hygiene programs in Nigeria have been structured. 

Research methodology. Too much useless information in chapter 2.1. It doesn't matter how many people were chosen or how many agreed. What is important is the final result - how many people participated in the study until the end.

Chapter 2.3. it will be more readable if these survey questions are bulleted. This paragraph is very difficult to read.

Results clearly presented and well discussed.

Summary chapter too extensive. Perhaps it would be worth pointing out what actions should be taken to improve the sanitary situation in Nigeria. This chapter should be shortened.

Author Response

Reviewer 2 Comments

Introduction. Shorten the paragraph on the role of higher education as an element of social development. Too general statements. What matters is, how academic sanitation and  hygiene programs in Nigeria have been structured. 

Reply to the Reviewer 2

Paragraph and the whole introduction reviewed and reduced

Reviewer 2 Comments

Research methodology. Too much useless information in chapter 2.1. It doesn't matter how many people were chosen or how many agreed. What is important is the final result - how many people participated in the study until the end.

Reply to the Reviewer 2

Method section reviewed and 2.1 corrected

Reviewer 2 Comments

Chapter 2.3. it will be more readable if these survey questions are bulleted. This paragraph is very difficult to read.

Reply to the Reviewer 2

Section 2.3 reviewed as suggested

Reviewer 2 Comments

Results clearly presented and well discussed

Reply to the Reviewer 2

Thank you

Reviewer 2 Comments

Summary chapter too extensive. Perhaps it would be worth pointing out what actions should be taken to improve the sanitary situation in Nigeria. This chapter should be shortened.

Reply to the Reviewer 2

  • Summary section reviewed and properly presented with recommendations properly delineated.
  • The focus of the paper is sanitation and hygiene education and not particularly the sanitation situation in Nigeria, that is why the recommendation proposes a transformative pathway to improve sanitation and hygiene education processes in the country.

Reviewer 3 Report

Comments and Suggestions for Authors

General comments:

Please ensure the figures' font matches the main text font.

Abstract

When mentioning "limited understanding of the concepts of contemporary issues," it would be beneficial to specify which concepts or issues. This will help international authors grasp the scale and scope of the problem. The methodology section might benefit from more detail explaining why a multilevel mixed method concurrent study was chosen and how it was implemented. The current explanation is unclear.

 Introduction

Please make sure all acronyms (e.g., HE, SH, HEIs) are first used and used consistently.

While the challenges are stated (e.g., outdated curricula, dissatisfaction among clients/service users), tightening how these problems are presented and relating them directly to the study's objectives will sharpen your introduction.

The study aims are detailed in the concluding part; having a dedicated section that briefly states the study objectives might offer better emphasis and clarity.

Materials and Methods:

It would be helpful to provide references where the methodology mentioned in 2.4.1 is applied.

Comments on the Quality of English Language

Please fix the line spacing, preposition, and spelling errors. Please proof read it thoroughly.

Author Response

Reviewer 3 Comments

Abstract: When mentioning "limited understanding of the concepts of contemporary issues," it would be beneficial to specify which concepts or issues. This will help international authors grasp the scale and scope of the problem. The methodology section might benefit from more detail explaining why a multilevel mixed method concurrent study was chosen and how it was implemented. The current explanation is unclear.

Reply to the Reviewer 3

Considered and effected

Reviewer 3 Comments

 Introduction: Please make sure all acronyms (e.g., HE, SH, HEIs) are first used and used consistently.

Reply to the Reviewer 3

Considered and effected

Reviewer 3 Comments

Introduction: While the challenges are stated (e.g., outdated curricula, dissatisfaction among clients/service users), tightening how these problems are presented and relating them directly to the study's objectives will sharpen your introduction.

Reply to the Reviewer 3

Considered and effected

Reviewer 3 Comments

Introduction: The study aims are detailed in the concluding part; having a dedicated section that briefly states the study objectives might offer better emphasis and clarity.

Reply to the Reviewer 3

Considered and effected

Reviewer 3 Comments

Materials and Methods: It would be helpful to provide references where the methodology mentioned in 2.4.1 is applied.

Reply to the Reviewer 3

References are included in 2.4.1

Reviewer 3 Comments

Please fix the line spacing, preposition, and spelling errors. Please proof read it thoroughly.

Reply to the Reviewer 3

Considered and effected

Round 2

Reviewer 1 Report

Comments and Suggestions for Authors

On account of the manuscript HYGIENE-2652955R1, entitled “Multi-level Perceptions on Higher Education Development for Sanitation and Hygiene Management in Nigeria” by Peter Emmanuel Cookey et al., the author revised the manuscript appropriately according to the Reviewers comments. After careful consideration, I made a decision that the manuscript is acceptable for publication in its present form.

Author Response

We thank you for helping to improve the quality of our manuscript
